# Accurate Split Learning on Noisy Signals

## Abstract

Noise injection is applied in Split Learning to address privacy concerns about data leakage. Previous works protect Split Learning by adding noise to the intermediate results during the forward pass. Unfortunately, noisy signals significantly degrade the accuracy of Split Learning training. This paper focuses on improving the training accuracy of Split Learning over noisy signals while protecting training data from reconstruction attacks. We propose two denoising techniques, namely scaling and random masking. Our theoretical results show that both of our denoising techniques accurately estimate the intermediate variables during the forward pass of Split Learning. Moreover, our experiments with deep neural networks demonstrate that the proposed denoising approaches allow Split Learning to tolerate high noise levels while achieving almost the same accuracy as the noise-free baseline. Interestingly, we show that after applying our denoising techniques, the resultant network is more resilient against a state-of-the-art attack compared to the simple noise injection approach.

## 1 Introduction

Privacy concerns in various application domains, including finance, healthcare, and online commerce, limit the sharing of raw data required to train accurate deep neural networks (DNNs). Split Learning Gupta & Raskar (2018); Vepakomma et al. (2018a) has emerged as a solution that enables different parties to collaboratively learn a model, without explicitly sharing raw input data. Typically, in two-party Split Learning, the Split Neural Network (SplitNN) is divided between the *data owner*, a.k.a. the client, and the *label owner*, a.k.a. the server; see Figure 1 (a). During training, in the forward pass, the client forwards the intermediate results (IRs) (i.e., the neurons of the *cut layer*, the last layer in the client's part of the DNN) to the server. The server completes the forward pass, and during the backpropagation, it returns the gradients of the IRs to the client. Consequently, the client can train the joint model without revealing the private training data to the server.

Unfortunately, sharing only the IRs does not protect the raw data. The shared IRs contain considerable latent information about the data and can be used to stage powerful attacks, such as model inversion attack He et al. (2019a); Zhang et al. (2020); Erdogan et al. (2021), label inference attack Erdogan et al. (2021); Kariyappa & Qureshi (2021); Li et al. (2021), and hijacking attack Pasquini et al. (2021). Several works Titcombe et al. (2021); Abuadbba et al. (2020); Mireshghallah et al. (2020); Wang et al. (2018) attempt to mitigate that risk through adding a certain amount of noise to the IRs before sharing with the other party. However, one of the fundamental issues of adding noise is the trade-off between the trained model's quality and its susceptibility to an external attacker. While high levels of noise are favorable in making the training data private, the noise inevitably impacts the quality of the trained model Abuadbba et al. (2020); Wang et al. (2020a; 2021), that is, the accuracy; see Figures 1 (b) and (c) for an example. Thus, a fundamental question is: Can we improve the training accuracy of Split Learning under noise injection, without making the model vulnerable to data leakage?

We answer the above question affirmatively by applying a post-processing denoising layer on top of noise-injected IRs in the Split Learning process. Our intuition is that *the injected noise introduces an error during the forward pass, which is dominated by variance when the noise level is high*. As long as we can reduce the variance by using denoising techniques, the training quality should be improved. Such a post-processing layer will not impose any additional private information leakage as long as it does not interact with the original private data.

We list our contributions as follows:

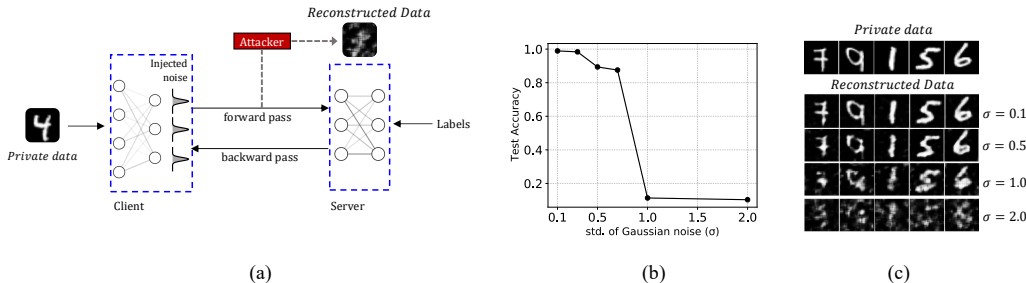

(a)           (b)           (c)

Figure 1: The trade-off between the security and training accuracy in noise-injected private Split Learning. (a) General schematic representation of two-party split learning, where the neural network is split between the client (owns the data) and server (owns the labels). Noise is injected at the client's side output to prevent private data leakage from the attacker. (b) Noise variance ($\sigma$) vs. Test accuracy in training a split CNN model on the MNIST dataset. As the variance of the injected noise level ($\sigma$) increases, the test accuracy drops. (c) Training data reconstruction by hijacking attack at different injected noise levels. The reconstruction capacity decreases as the injected noise level is increased.

**Contributions.** (*i*) We propose two denoising techniques (i.e., scaling and masking) to improve the training accuracy and stability of noise-injected SplitNN; see §3. (*ii*) Our theoretical investigation on a classification task in §3.2 shows that denoising can reduce the error caused by noise injection during the forward pass. (*iii*) In addition to improving the train model quality, we show that our modification to Split Learning with noise injection, followed by postprocessing, preserves the security guarantee in the entire training protocol; see §3.3. Quantifying the privacy aspect in our work is not the primary focus; it is an additional benefit. (*iv*) We validate our claims through extensive numerical experiments on synthetic and real data (i.e., *7 DNN models on 7 different datasets, including large-scale datasets, ImageNet1K Deng et al. (2009) and Amazon Reviews McAuley & Leskovec (2013)*) in §4. Moreover, (*v*) we find that our masking technique, in addition to denoising, also enhances the resilience of Split Learning against the state-of-the-art hijacking attack Pasquini et al. (2021); refer to §4.3.

## 2 RELATED WORK

**Federated Learning (FL) with noise injection.** Noise injection in vertical FL shares some similarities with noise-injected SplitNN since in both cases, the noise is injected on intermediate results during the forward pass. Existing works Wang et al. (2020b); Chen et al. (2020) propose to add Gaussian/Laplacian noise on participants' IRs to protect private training data or labels. Chen et al. (2020) only demonstrates the impact on training accuracy when some applications have a relatively low noise scale. The other framework Wang et al. (2020b) proposes a similar noise injection technique only for linear model collaborative learning, which is not directly applicable to general DNNs.

**Split Learning with noise injection.** Due to the vulnerability of SplitNN against model inversion attacks, Titcombe et al. (2021) proposed to apply differentially private noise injection on IRs during the inference time to prevent data reconstruction by the attacker. Shredder, proposed by Mireshghallah et al. (2020), adaptively generates a noise mask to minimize mutual information between input and intermediate data. However, these two methods only introduce noise injection during the inference time; thus, the privacy of training data is not preserved. Abuadbba et al. (2020) successfully applies noise to the IRs during the training to defend against model inversion attacks on one-dimensional ECG data; also, see Wu et al. (2023) for SpiltNN with differential privacy for integrated terrestrial and non-terrestrial Networks. It turns out that the noise has dramatically impacted the model's accuracy. Unlike previous works that only focus on the attack defense efficacy, we aim to improve the training accuracy with a significant noise level.

We provide an answer to split learning denoising by proposing two post-processing techniques (i.e., scaling and masking) to improve the accuracy and stability of the splitNN training when Gaussian noise is injected into the IRs. To our knowledge, we are the first to propose denoising techniques on the Gaussian noise-injected IRs to improve the training accuracy of SplitNN and theoretically show the privacy guarantee in this setup. In the following, for completeness, we mention some differential privacy (DP) techniques used in federated learning (FL) and related settings for completeness.

**Gaussian noise injections (GNIs)** are a family of regularization methods for DNN training through adding Gaussian noise on the activations or weights during the forward pass. It is similar to the noise

injection in SplitNN except for the following two aspects: (*i*) There is no requirement of bounded sensitivity in injection objects. (*ii*) The noise scale is usually set small to avoid negative impacts on training accuracy. The explicit regularization effects of GNIs are well investigated in Camuto et al. (2020); Li & Liu (2020); Lim et al. (2021), demonstrating better generalization for trained models over unseen data. In addition, GNIs can improve the robustness of DNNs against adversarial attacks or data perturbations Lim et al. (2021); He et al. (2019b). However, Camuto et al. (2021) also found that GNIs can introduce some implicit bias on gradient updates, which inevitably degrades the training accuracy.

**Denoising.** Adding the Gaussian noise and the denoising mechanisms in our work shares many similarities with the differential privacy (DP) and their post-processing that maintains the DP guarantee and often improves accuracy Zhu et al. (2022; 2021). Denoising for DP has been well adopted in the statistical estimation Hay et al. (2009; 2010); Nikolov et al. (2013); Bernstein et al. (2017), where they exploit some prior knowledge to design a data release mechanism with better DP utility. Recently, Balle & Wang (2018) proposed an optimal denoising technique for Gaussian mechanism, where given $y \sim \mathcal{N}\left(f(x), \sigma^2 I\right)$ and their target is to find a postprocessing function $g$ such that $g(y)$ is closer to $f(x)$ than $y$. This is substantially different from SplitNN as there are subsequent layers on top of the Gaussian mechanism in the training process. Nasr & Shokri (2020) has also investigated using scaling as a denoising technique to improve the DP utility for DP-SGD Abadi et al. (2016). However, the authors scale up/down the noisy gradients based on the "usefulness" of gradients, while we utilize scaling to minimize the estimation error of the noisy neural network outputs. Wang et al. (2020a) showed that adding Laplacian smoothing on Gaussian noise-injected gradients can improve the utility of DP-SGD. While Ligett et al. (2017) proposed a general noise reduction framework, Lecuyer et al. (2019) proposed PiXel-DP, an adversarial defense mechanism, scalable to diverse large networks and datasets.

## 3 THEORETICAL GUARANTEE

**Notations.** By $[n]$ we denote the set of $n$ natural numbers $\{1, 2, \cdots, n\}$. By $x_i$, we denote the $i^{\text{th}}$ component of vector $x$, while $A_{ij}$ denote the $(i, j)^{\text{th}}$ component of a matrix, $A$. We use $\|x\|_2$ and $\|A\|_F$ to denote the $\ell_2$ and the Frobenius norms of a vector $x$ and a matrix $A$, respectively.

**Problem setup.** Let $D$ be the training dataset with $N$ elements, $\{(X_i, y_i^\star)\}_{i=1}^N$, drawn i.i.d. from some distribution, $P(\mathcal{X}, \mathcal{Y})$, where $X_i \in \mathbb{R}^d$ is the input feature vector, and $y_i^\star$ is the corresponding ground-truth label. We consider a SplitNN with an output vector in $\mathbb{R}^m$. The network is divided between the client and the server, where the server network consists of several DNN layers and the output loss function. In our experiments, we show different split configurations; however, in our theoretical analysis, we have one fully connected (FC) layer after the cut layer as the final layer.

Let $X \in \mathbb{R}^n$ be the vector from the client-side cut layer, and $M \in \mathbb{R}^{m \times n}$ be the weight matrix of the FC layer on the server side. At each iteration during training, the original split network processes a minibatch of training samples to calculate the loss and, during backpropagation, updates the gradients. We follow this formalization in our theoretical analysis.

We consider the GNIs to protect the vector $X$. For Laplacian noise, see §B.1. Let the perturbed vector, $\tilde{X} \in \mathbb{R}^n$ follow the model: $\tilde{X} = X + Z$, where $Z_i \sim \mathcal{N}(0, \sigma^2)$, chosen from a zero mean Gaussian distribution with standard deviation $\sigma \in \mathbb{R}^+$. Then, we apply a post-processing function $h_D(\cdot) : \mathbb{R}^n \to \mathbb{R}^n$ on $\tilde{X}$ before forwarding it to the server side. To bound the magnitude of $X$, we use `tanh` as the activation function.

If we set the cut layer at an arbitrary $i$-th layer in an $L$-layer DNN, along with the final loss function used for DNN training, the theoretical analysis would become less tractable. Therefore, in our setup, we set $i = L - 1$ to illustrate the main ideas. We formalize two possible cases in the forward pass to the output layer: (*i*) a linear layer, $\Phi = I_m$, an identity map, and we quantify $\mathbb{E}\|MX - Mh_D(\tilde{X})\|_2^2$ against $\mathbb{E}\|MX - M\tilde{X}\|_2^2$; see §3.1, and (*ii*) a linear layer with a nonlinear activation function (Softmax), $\Phi = s$, nonlinear loss function (negative log), $\mathcal{L}_{LL}$, and we quantify $\mathbb{E}|\mathcal{L}_{LL}(y^\star, s(MX)) - \mathcal{L}_{LL}(y^\star, s(Mh_D(\tilde{X})))|$ against $\mathbb{E}|\mathcal{L}_{LL}(y^\star, s(MX)) - \mathcal{L}_{LL}(y^\star, s(M\tilde{X}))|$, where $y^\star$ is the true label vector; see §3.2. In both cases, $h_D$ is a post-processing mechanism that is used to improve the training quality, resulting in some "denoising" effect; the signal-to-noise ratio is not the measure of "denoising" in our context.

Our primary focus is on nonlinear classification tasks, although for a better understanding, denoising the linear layer is important, which can be viewed as a regression task. Our theorems are proxies to measure whether the postprocessing would allow better server accuracy. We do not include an iteration counter on $M, X$, or $\tilde{X}$; we are examining the single forward pass. We use a random masking operator, $R_p$, and a scaling operator, $S_\alpha$, as $h_D$; see §A.1 for their definitions.

### 3.1 A LINEAR LAYER

With the above setup, to easily explain our ideas, we start with a neural network performing simple regression. Although this is not our primary focus, we believe this section provides a better understanding in a simple setting. Because for the $\ell_2$-regression task, no nonlinear activation function is required, the problem is much simpler. That is, $y := MX$ is the prediction of the output layer, and it does not involve any non-linearity. Theorem A.1 describes results for a fully-connected DNN with an $\ell_2$-regression task. Additionally, it explains how the scaling and masking parameters, $\alpha$ and $p$, respectively, are related to the noise scale $\sigma$, while denoising the output of a linear layer of a DNN for a given $M$ and $X$. We calculate the expected test error, $\mathbb{E}\|MX - Mh_D(\tilde{X})\|_2^2$, where $h_D$ is $R_p$ or $S_\alpha$, and compare it against $\mathbb{E}\|MX - M\tilde{X}\|_2^2$. Due to limited space, we put the result in §A.2.

### 3.2 NONLINEAR LOSS FUNCTION FOR CLASSIFICATION TASK

For a vector, $z \in \mathbb{R}^m$, denote $s : \mathbb{R}^m \to (0,1)^m$ as the softmax function, and $\mathcal{L}_{LL}(y^\star, s(z))$ as the negative log loss function, where $y^\star$ is the true label vector; see definition in §A.1. In what follows, we show that for both masking and scaling operators, under certain conditions on the noise level, $\sigma$, it is possible to find parameters $p$ and $\alpha$, respectively, such that, by using any of these operations, we incur a lower deviation in the loss value than using the noise injection alone when compared to the loss of the original SplitNN.

**Masking operation.** Quantifying $\mathcal{L}_{LL}(y^\star, s(MR_p(\tilde{X}))$ and $\mathcal{L}_{LL}(y^\star, s(M\tilde{X}))$ are critical as they involve randomness from the masking operator and the Gaussian noise. We require several intermediate results to prove the main result in Theorem 3.1. We state and prove them in the § A.3.1.

We want to show that by using a random mask over a noise-injected layer, we incur a lower deviation in the loss value than using the noise injection alone when compared to the loss, $\mathcal{L}_{LL}(y^\star, s(MX))$, of the original SplitNN under certain conditions. That is, we want to compare the quantities $\mathbb{E}|\mathcal{L}_{LL}(y^\star, s(MX)) - \mathcal{L}_{LL}(y^\star, s(MR_p(\tilde{X})))|$ and $\mathbb{E}|\mathcal{L}_{LL}(y^\star, s(MX)) - \mathcal{L}_{LL}(y^\star, s(M\tilde{X}))|$. The following result formalizes this.

**Theorem 3.1.** *With the notations above, for classification problems, assume that $n \geq (MX)_i$ for $i = 1, 2, ..., m$. Then, if $\sigma$ is large enough, there is some $\delta \in (0, 1)$ such that for $p \in (\delta, 1]$,*

$$\mathbb{E}|\mathcal{L}_{LL}(y^\star, s(MX)) - \mathcal{L}_{LL}(y^\star, s(MR_p(\tilde{X})))| \leq \mathbb{E}|\mathcal{L}_{LL}(y^\star, s(MX)) - \mathcal{L}_{LL}(y^\star, s(M\tilde{X}))|.$$

The assumption, $n \geq (MX)_i$ is technical and can be easily satisfied in practice, which requires the input dimension from the SplitNN to be wide enough. We will pause here and provide a sketch of proof of Theorem 3.1. Because the original SplitNN always produces the least loss, the expressions in absolute values in the inequality above are non-positive, and so we need only to verify that for all $X$, $\mathbb{E}\mathcal{L}_{LL}(y^\star, s(MX)) - \mathcal{L}_{LL}(y^\star, s(M\tilde{X})) \leq \mathbb{E}\mathcal{L}_{LL}(y^\star, s(MX)) - \mathcal{L}_{LL}(y^\star, s(MR_p(\tilde{X})))$. By the definitions of softmax and negative log loss, we have

$$\mathcal{L}_{LL}(y^\star, s(MR_p(\tilde{X}))) = -(MR_p(\tilde{X}))_{i^\star} + \log\left(\sum_{i=1}^m e^{(MR_p(\tilde{X}))_i}\right), \quad (1)$$

where $i^\star$ is the location of true label in $y^\star$. For fixed $M$ and $\tilde{X}$, (1) is a function of $p$ for $p \in (0, 1]$. Denote $\mathcal{F}(p) := \mathbb{E}\mathcal{L}_{LL}(y^\star, s(MR_p(\tilde{X})))$, and consequently, $\mathcal{F}(1) = \mathbb{E}\mathcal{L}_{LL}(y^\star, s(M\tilde{X}))$; see Remark A.6. By using Lemma A.7 on (1), we can approximate $\mathcal{F}(p)$ by

$$\mathcal{F}(p) \approx -p(MX)_i + \log\left(\mathbb{E}\sum_{i=1}^m e^{(MR_p(\tilde{X}))_i}\right) - \frac{\mathrm{Var}\left(\sum_{i=1}^m e^{(MR_p(\tilde{X}))_i}\right)}{2\left(\mathbb{E}\left(\sum_{i=1}^m e^{(MR_p(\tilde{X}))_i}\right)\right)^2}. \quad (2)$$

We want to show that $\mathcal{F}(p) \leq \mathcal{F}(1)$ when $p \in (\delta, 1)$, for some $\delta \geq 0$. By using Lemma A.5 in (2) and differentiating with respect to $p$, we can show, $\mathcal{F}'(1) \geq 0$. This would imply that $\mathcal{F}(p)$ is an increasing function of $p \in (\delta, 1]$, for some $\delta \in (0, 1)$. This gives us $\mathbb{E}\mathcal{L}_{LL}(y^\star, s(MR_p(\tilde{X}))) \leq \mathbb{E}\mathcal{L}_{LL}(y^\star, s(M\tilde{X}))$, and §A.3.1 concludes the proof of Theorem 3.1.

**Scaling operation.** Similarly, by using scaling over $\tilde{X}$, under certain conditions on the noise level, $\sigma$, we obtain a lower deviation in the loss than using the noise injection alone when compared to the loss of the original SplitNN. Let $\tilde{\mathcal{L}}_{S_\alpha} := \mathcal{L}_{LL}(y^\star, s(MS_\alpha(\tilde{X}))$. We state the result in Theorem 3.2; see §A.3.2 for a sketch of the proof.

**Theorem 3.2.** *With the notations above, for classification problems, if* $\sigma^2 \geq \max_{i,i^\star} \frac{\sum_{k=1}^n (m_{i^\star k} - m_{ik})x_k}{\sum_{k=1}^n m_{ik}^2}$, *for* $i = 1, 2, ..., m$, *then there exists a* $\delta' \in (0, 1)$ *such that* $\mathbb{E}|\mathcal{L}_{LL}(y^\star, s(MX)) - \tilde{\mathcal{L}}_{S_\alpha}| \leq \mathbb{E}|\mathcal{L}_{LL}(y^\star, s(MX)) - \mathcal{L}_{LL}(y^\star, s(M\tilde{X}))|, \text{for } \alpha \in (1, \frac{1}{\delta'}]$.

For concentration of the errors in Theorem A.1 and Theorem 3.1; see §A.3.3.

### 3.3 DIFFERENTIAL PRIVACY (DP) PRESERVATION

Recent works show that Split Learning with Laplacian or Gaussian noise injection at the cut layer is resilient to attacks Abuadbba et al. (2020); Wu et al. (2023). In particular, DP (see Definition in §A.4) could be used to describe the privacy guarantee. In this sub-section, we explain how our modification (scaling or masking) to Split Learning with noise injection (such as GNI) preserves DP. The key to our argument is to view our modification as a post-processing of a DP mechanism. Then by the immunity of DP to post-processing, we can conclude that DP will be preserved.

Next, in light of our results in §3.1 and §3.2, we recall some terminologies needed for our discussion. For more details, see Dwork et al. (2014); Xiang et al. (2019); Abadi et al. (2016).

Let $\mathcal{D}$ be a collection of databases. $D \subset \mathcal{D}$ and $D' \subset \mathcal{D}$ be two neighboring training datasets, that is, $D' = D \pm \{X\}$. In this case, we write $d(D', D) = 1$. For a function $V : \mathcal{D} \to \mathbb{R}^m$, define a randomized mechanism, $\mathcal{K} : \mathcal{D} \to \mathcal{O}$ such that, for $D \in \mathcal{D}$,

$$\mathcal{K}(D) := V(D) + Z, \tag{3}$$

where $Z \in \mathbb{R}^m$ is a random variable/vector with probability density function, $p(z)$.

To ensure $\mathcal{K}$ is $(\epsilon, \delta)$- DP, one must require some condition on the density function $p(z)$ of $Z$. For example, if $p(z)$ is Laplacian or Gaussian density functions with their variances satisfying some lower bounds, then $\mathcal{K}$ is DP. But, before we recall the result, we need one more important concept.

Define the *sensitivity* of $V$ as

$$\Delta := \sup_{D, D', d(D,D')=1} \|V(D) - V(D')\|_2. \tag{4}$$

The following Theorem by Dwork et al. (2014) says, in the case when $m = 1$, if $z \sim \mathcal{N}(0, \sigma^2)$, then the random mechanism $\mathcal{K}$, as defined in (3) is DP as long as $\sigma$ is large enough.

**Theorem 3.3.** *(Dwork et al., 2014, Theorem 3.22) Let $\epsilon \in (0, 1)$ be arbitrary. For $c^2 > 2\ln(1.25/\delta)$, the mechanism in (3) with parameter $\sigma \geq c\Delta/\epsilon$ is $(\epsilon, \delta)$-differentially private.*

In Theorem 3.3, we note that the noise level $\sigma$ is directly proportional to the sensitivity $\Delta$ and inversely proportional to the privacy bound $\epsilon$.

This result will give us the DP of GNI for Split Learning (Lecuyer et al., 2019, §III.B), which suggested putting the cut layer early in the network, where bounding the sensitivity is easier. The next result allows us to see that both scaling and masking proposed in this work will preserve DP.

**Theorem 3.4.** *(Dwork et al., 2014, Proposition 2.1) Let $\mathcal{K}$ be a randomized mechanism which is $(\epsilon, \delta)$-DP. If $g : \mathbb{R}^m \to \mathbb{R}^d$ is a deterministic or randomized mapping, then $g \circ \mathcal{K}$ is also $(\epsilon, \delta)$-DP.*

**Impact on backpropagation.** Now, we discuss the induced change during the backpropagation. In DNN training, backpropagation is used to evaluate the gradients, which will be used to update the estimation of parameters. As the popular DP procedure applied to DNN is DP-SGD Abadi et al.

(2016), we indicate how our scaling and masking would "alter" the gradients that could be viewed as an approximate DP-SGD. Let $g^t$ be the gradient and $g_C^t$ denote the clipped version of $g^t$ during the $t$-th iteration. DP-SGD would add Gaussian noise $Z \sim \mathcal{N}(0, \sigma I)$ to $g_C^t$. What happened to our case is that starting from the beginning of the backpropagation, we will evaluate gradients at the "perturbed" values (due to the GNI and our modification using scaling or masking from the cut layer) and these values will be used to propagate backward at each layer to obtained $\tilde{g}_C^t$, a perturbed version of $g^t$ and hence $g_C^t$. Thus, we can put this into the framework of (3) with $\mathcal{K}(D) = \tilde{g}_C^t(D)$, $V(D) = g_C^t(D)$, and write

$$\tilde{g}_C^t(D) = g_C^t(D) + Z,$$

where $Z$ is a random variable/vector due to the randomness of GNI and masking (using Bernoulli distribution). Although $Z$ itself may not be Gaussian, it can be shown that $Z = g(W)$ where $g$ is a sum of products of compositions of affine transformations and activation functions, and $W$ is a random vector of independent Gaussian and Bernoulli variables. Let $p(z)$ denote the "density" of $Z$. Denote the sensitivity of $g_C^t(\cdot)$ by $\Delta_g$. Define

$$S = \{w : p(w) > e^\epsilon p(w + \Delta_g)\}.$$

We can establish the following theorem, whose proof is given in the §A.13.

**Theorem 3.5.** *Let $\delta > 0$ and recall that $Z$ is a random variable with probability density $p(z)$. Then, there exists an $\alpha > 0$ such that, for $\Delta_g \leq \alpha$,*

$$Pr(Z \in S) < \delta,$$

*where $P[\cdot]$ refers to the probability associated with the random variable. Furthermore, when $\Delta_g \leq \alpha$, $\tilde{g}_C^t$ is $(\epsilon, \delta)$-DP.*

Note that $\Delta_g$ is small for large dataset (that is, when $|D|$ is large). So, we can make the sensitivity small by requiring large training datasets.

The above explains DP in one iteration of the training data. If we consider a total of $T$ training iterations, we can use the advanced composition theorem, Theorem A.10 to guarantee, the mechanism is $(\epsilon\sqrt{2T \ln(1/\delta')} + T\epsilon(e^\epsilon - 1), T\delta + \delta')$-DP for all $\delta' > 0$.

We can give a similar guarantee when the noise mechanism is Laplace. Also, note that, if in each training iteration, $g_{C,B}^t$ represents the clipped gradient calculated over a minibatch of size $B$ taken from large enough databases, then for sensitivity, $\Delta_{g,B}$ small enough, the DP guarantee for $g_{C,B}^t$ holds following the same argument as above; see Xiang et al. (2019), and our discussion in §A.4.

In our experiments, we use the regular backpropagation formula for our noisy split network training; see Proposition A.11. Noise injection and postprocessing in the forward pass perturb the gradients during the backward pass, but without adding any explicit noise to them in each iteration. However, this is not the same as GNIs to the gradient as in DP-SGD Abadi et al. (2016)—The noise, in our case, (*i*) is a more general, data-adapted random variable than Gaussian, and that (*ii*) it is generated over the DNN architecture, not a user-specified Gaussian noise.

# 4 EXPERIMENTAL EVALUATION

In §4.1, we validate our theoretical claims through simulation on synthetic data, §4.2 shows results on DNNs performing machine learning tasks, Section §4.3 shows improved data privacy results.

## 4.1 SIMULATION

**Setup.** The following numerical simulations verify the results of Theorem A.1 and 3.1. Since $X \in [-1, 1]$ is the output of `tanh` function, and $M$ is usually randomly initialized around 0 in the actual training, we sample the entries of $X$ and $M$ from a uniform distribution on $[-1, 1]$ in our simulation. The MSE of masking corresponds to $\mathbb{E}\|f(X) - f(R_p(\tilde{X}))\|_2^2$ for all different functions ($f_1, f_2, f_2'$ in Figure 2), where $R_p$ can be replaced by $S_\alpha$ for scaling. Moreover, when $p = 1$ and $\lambda = \frac{1}{\alpha} = 1$, masking and scaling are ineffective thus the respective MSEs are considered baseline MSEs. In Figure 2, for each plot, we draw a line parallel to the X-axis from these baseline MSEs. The expectations are calculated by taking the average on $k$ simulation results, where $k = 1000$.

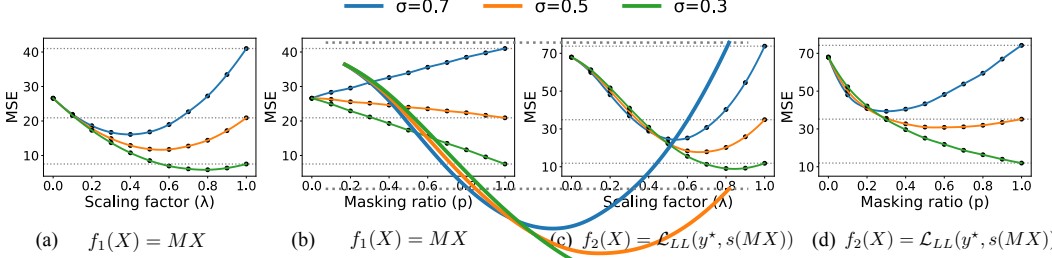

(a)     $f_1(X) = MX$     (b)     $f_1(X) = MX$     (c)     $f_2(X) = \mathcal{L}_{LL}(y^\star, s(MX))$     (d)     $f_2(X) = \mathcal{L}_{LL}(y^\star, s(MX))$

Figure 2: Simulation of how scaling factor ($\lambda = \frac{1}{\alpha}$) and masking ratio ($p$) influence the estimation error (MSE) under different noise levels ($\sigma$) for linear ($f_1$) and nonlinear functions ($f_2, f_2'$).

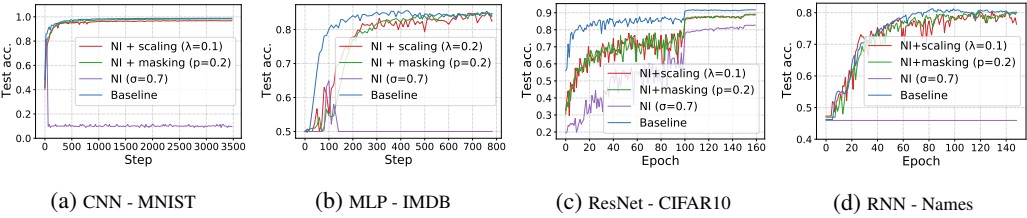

(a) CNN - MNIST     (b) MLP - IMDB     (c) ResNet - CIFAR10     (d) RNN - Names

Figure 3: Test accuracy of SplitNN training with noise injection (NI) only and noise injection (NI) plus denoising (i.e. masking or scaling) in different training tasks. All models are split with one FC layer on the server side. ($\sigma$: noise level, $p$: masking ratio, $\lambda$: scaling factor $\frac{1}{\alpha}$)

**Scaling simulation.** In Figure 2 (a), each curve corresponds to a different noise scale, $\sigma$. By decreasing the scaling factor, $\lambda$ for each $\sigma$, the MSE first decreases from the baseline to a minimum then increases, indicating an optimal $\lambda$ for each $\sigma$. The NASC condition in Theorem A.1 (*ii*) also infers that. For fixed $M, X$, this condition implies it is possible to find a smaller $\lambda$ when $\sigma$ is large. We make similar observations for the nonlinear case; see Figure 2 (c).

**Masking simulation.** Figure 2 (b) shows that by decreasing the masking ratio, $p$, the MSE does not necessarily become smaller unless $\sigma$ is large enough. This verifies the claim of Theorem A.1(*i*). More importantly, there is an *almost linear* relationship between MSE and the masking ratio as $p \to 1$. This coincides with the expression of MSE with masking given in equation 9; see Appendix. We hypothesize that while both $X, M$ are drawn from Uniform distribution, the coefficient of $p^2$ might become negligible. Hence, the coefficient of the linear term, $p$, which can be positive or negative depending on the noise scale $\sigma$, dominates the MSE. Results from Figure 2(d), with the nonlinear loss, reflect Theorem 3.1: $\sigma^2$ must be large for the *improvement to be possible*. If $\sigma$ is too small (MSE curve for $\sigma = 0.3$), the masking does not work; the larger the $\sigma$, the more improvements one can expect by using masking. Moreover, when $\sigma$ is large enough, there exists an $p \in (\delta, 1)$, for some $\delta \geq 0$ such that masking incurs a lower MSE than the baseline. This indicates that optimal denoising is possible by using masking for large noise. Nevertheless, for the same noise level, the MSE of the optimal denoising of masking is always larger than that of scaling. We provide the backward pass simulations in Figure 6 in §B. Figure 7 shows the simulation results for the Laplace mechanism, and they are discussed in detail in §B.1 along with experimental results in Table 3.

**Takeaway message.** Figures 2 (a), (c), and 6 (a) indicate that regardless of the noise scale, $\sigma$, it is possible to find a scaling factor such that using scaling over a noise-injected SplitNN incurs a lower MSE than the baseline. However, this is not always the case for the masking operator—$\sigma$ must be significant for rendering the improvement. In practice, we witness masking performs better than scaling in terms of improved accuracy, parameter-tuning, and attack defense; see §4.2 and 4.3.

## 4.2 DNN Experiments

**Datasets and models.** We adopt the benchmarks from the popular `Pytorch` library `Opacus` Yousefpour et al. (2021) with the Split Learning paradigm. It contains image classification tasks (on MNIST LeCun et al. (1998), CIFAR-10, CIFAR-100 Krizhevsky et al. (2009), and ImageNet1K Deng et al. (2009)), recommendation task (movie review prediction on IMDB Maas et al. (2011)),

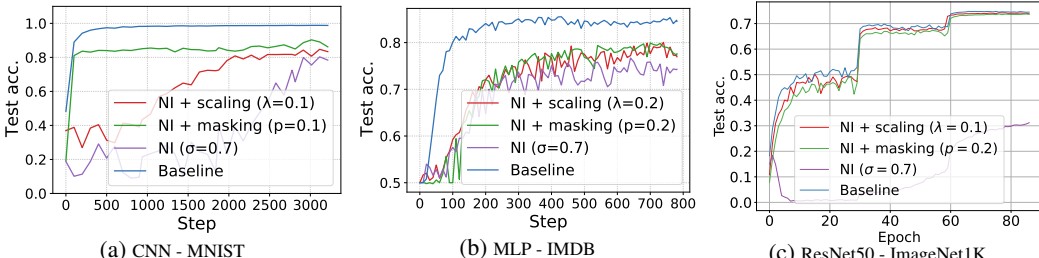

Figure 4: Test accuracy of SplitNN training with noise injection (NI) only and noise injection (NI) plus denoising (i.e. masking or scaling); *(a)* CNN on MNIST, *(b)* MLP on IMDB, *(c)* ResNet50 on ImageNet1K.

language modeling task (name classification Robertson (2023)), and sentiment analysis (Amazon reviews McAuley & Leskovec (2013)). All training hyperparameters are configured as default to maintain a fair comparison; see Table 2 in §B.

**Setup and implementation.** We split the models before fully connected (FC) layers, with a variety in the number of FC layers allocated at the server side; see Table 1. The size of the split layer varies from 16 to 12544. We use `tanh` activation function to bound the client's output in $[-1, 1]$. Then Gaussian noise is injected on the `tanh` layer, with noise scale, $\sigma$, the standard deviation of the Gaussian distribution. We implement both denoising techniques as a post-processing layer on top of the noise injection process. The ratio $p \in (0, 1)$ describes the percentage of the elements kept through masking. The scaling factor $\lambda = \frac{1}{\alpha} \in (0, 1)$ is used to scale down the tensor values. The overall computation paradigm is outlined in Table 9 in §B.6.

**Denoising performance.** We demonstrate the effectiveness of the denoising techniques in various SplitNN training tasks. In Figure 3 and 4, we compare baseline SplitNN, noise-injected SplitNN, and noise-injected SplitNN with the scaling or masking denoising in 2 different split settings. The noise level, $\sigma$ is calibrated to a relatively high such that the training accuracy of SplitNN suffers from the noise injection. Both scaling and masking are optimized by parameter tuning on the scaling factor, $\lambda$, and masking ratio, $p$; see Table 10 in §B. When models are split at the last FC layer, e.g., in Figure 3 (a)(b)(d), once we inject a large noise ($\sigma = 0.7$), the overall training convergence is severely impacted so that the test accuracy is barely increased during the training. In Figure 4, the training is more robust under high noise injection because the size of the splitting layer is much larger than the one in previous settings. Usually, high-dimensional data can better tolerate noise perturbation since it carries more information. After applying the scaling or masking and fine-tuning some hyperparameters, the training convergence vastly improves. In most cases, e.g., Figure 3(a)(b)(d), the improved accuracy due to masking is comparable with the baseline. However, in Figure 4, with scaling and masking, the test accuracy can not achieve the baseline level. This is possible because, by allocating more layers on the server side, the client's noisy IRs will also impact more layers during the forward computation. We also notice that in Figure 3 (c) the noise injected training on CIFAR-10 performs much better than other tasks even the split layer size is relatively small. This is due to its unique default parameter setup such as weight decay and learning rate, which we will explain next.

**Denoising vs. Hyperparameter tuning.** To better understand the difference between denoising and traditional hyperparameter tuning, we evaluate the MNIST image classification task by fine-tuning the learning rate (lr), weight decay, dropout, masking ratio, and scaling factor under high-level noise injection. We present the accuracy results in §B in Table 4. Full training curves are available in §B in Figure 8. We change the lr from 0.1 to 0.001 and find that a smaller lr indeed improves the training stability under a large noise injection. Weight decay, as a popular regularization method in DNN training, can be used to avoid over-fitting on noisy signals. We find that only a heavy weight decay ($\gamma = 0.2, 0.4$) can help stabilize the training convergence till the end. However, a heavy weight decay sacrifices the convergence speed and fails to reach the baseline accuracy. Scaling can only improve the convergence at the beginning of the training, and none of them manage to maintain the convergence till the end. This implies an inherent training stability issue with noise injection, which cannot be alleviated by pure denoising. Therefore, we combine scaling with weight decay and find that a small weight decay ($\gamma = 0.01$) is sufficient to stabilize the training. On the contrary, the optimization of masking does not need weight decay. It can almost achieve the baseline convergence rate once the ratio $p$ is properly tuned. Although there is a similarity between random masking and the dropout technique Srivastava et al. (2014), simply using dropout does not provide enough stability

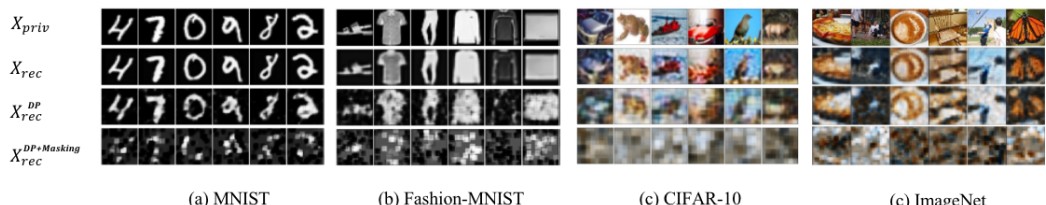

| $X_{priv}$ | | | |
| $X_{rec}$ | | | |
| $X_{rec}^{DP}$ | | | |
| $X_{rec}^{DP+Masking}$ | | | |
| (a) MNIST | (b) Fashion-MNIST | (c) CIFAR-10 | (c) ImageNet |

Figure 5: Private training data reconstruction by FSHA attack in Split Learning on MNIST and ImageNet. In all cases, $X_{priv}$: the original training data, i.e. ground truth; $X_{rec}$: FSHA on plain-text SplitNN; $X_{rec(N)}$: FSHA on SplitNN with noise injection (NI) ($\sigma = 0.7$); $X_{rec(S)}$: FSHA on SplitNN with NI and scaling ($\sigma = 0.7, \lambda = 0.2$); $X_{rec(M)}$: FSHA on SplitNN with NI and masking ($\sigma = 0.7, p = 0.2$). Models are split with one FC layer on the server side.

for the training, regardless of the dropout ratio. Both denoising techniques achieve significantly better training quality than standalone hyperparameter tuning. In §B.2, Table 5, we provide initial results of ResNet-18 on the CIFAR-100; our denoising techniques achieve higher accuracy than simple learning rate tuning. In Table 6 in §B.3, we demonstrate that the scaling postprocessing performs well even when Adam's updating rule eliminates gradient scale impact. See limitations of the proposed approach in §C. In Table 7, we show the performance on the large-scale datasets. In addition, we discuss the different layers of the networks to split in §B.5 along with results in Table 8.

### 4.3 ATTACK DEFENSE

**Setup.** We demonstrate how the random masking technique can improve data privacy in defense against the recent feature-space hijacking attack (FSHA) Pasquini et al. (2021) in Split Learning. FSHA hijacks the client's learning process from the server side during the training and performs the data reconstruction once the client's output feature is learned; see details of threat model in §B.7. We evaluate the attack performance with 2 models and 4 publicly available datasets—CNN for MNIST and Fashion-MNIST, ResNet for CIFAR-10 and ImageNet. See the model configuration in Table 1. We compare the attack performance by visualizing the reconstructed private data between FSHA attacks on plain-text SplitNN, noise-injected SplitNN, and noise-injected SplitNN with masking or scaling. We focus on the case where only one FC layer is on the server because it is more resilient against data reconstruction attacks, see discussion in Figure 9 in §B.

**Results.** Figure 5 shows that the original FSHA can reconstruct the private data with very high accuracy for MNIST but only keeps the original images' appearance for ImageNet. This is consistent with the attack performance in Pasquini et al. (2021)—attack on low-entropy images usually requires less effort and can produce a high-quality reconstruction. Next, we apply noise injection to the intermediate results and conduct data reconstruction on the perturbed data by FSHA. We observe that for MNIST, the digits on the reconstructed image are recognizable. For more complex and color image datasets (i.e. ImageNet), although noise can hide the details in the images, we can still relate the constructed image with the original one by looking at the outline or the background color. Lastly, when we combine noise injection with masking, the reconstructed images are fully damaged, and thus, the data security is greatly enhanced. While the scaling technique has almost no effect during the reconstruction attack, no matter how we set the scaling factor. See Figures 10-12 in §B for results with different $\sigma, p, \lambda$ and other datasets. Improving the privacy accounting for split learning is not the primary focus of this work, instead, we want to show how denoising can improve noisy SplitNN accuracy. Nevertheless, we provide the privacy bounds for one single forward pass during the SplitNN training in Table 11.

## 5 CONCLUSION

We propose scaling and masking as denoising techniques to achieve accurate Split Learning on noisy signals. We show theoretically and empirically that denoising helps achieve more accurate intermediate outputs in DNN training under noise injection that significantly improves the stability and accuracy of Split Learning. Additionally, we show that the masking technique can provide better security enhancement than scaling against powerful attacks. Although in theory, scaling has better denoising efficacy, masking is likely to show better accuracy improvement due to its easier parameter tuning. Finally, we demonstrate the possibility of co-optimization of denoising and attack defense.

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

# A  THEORETICAL GUARANTEE

First, we will start with the definition of softmax and negative log loss functions used for nonlinear classification.

## A.1  DEFINITIONS

**Random masking operator.**  Let $R_p$ be a random matrix of 1's and 0's with identical and independently distributed entries, $(R_p)_{ij} \sim \text{Bernoulli}(p)$. Denote the support set, $\Omega_p \subset [m] \times [n]$ of $R_p$ as $\Omega_p := \{(i,j)|(R_p)_{ij} = 1\}$. Based on this, for a matrix, $A \in \mathbb{R}^{m \times n}$,

$$(\text{R}_p[A])_{ij} = \begin{cases} A_{ij} & : i \in \Omega_p, \\ 0 & : \text{otherwise.} \end{cases}$$

From the definition, $R_p$ is linear and is a projection operator, that is, $R_p^2 = R_p$.

**Scaling operator.**  For a matrix $A \in \mathbb{R}^{m \times n}$ and $\alpha > 1$, denote the element-wise scaling operator, $S_\alpha(\cdot) : \mathbb{R}^{m \times n} \to \mathbb{R}^{m \times n}$, as $S_\alpha(A) = \frac{1}{\alpha}A$. Unlike the random masking, the scaling operator, $S_\alpha$ has no randomness.

**Softmax and negative log loss.**  Let $m$ be the number of classes. For a vector, $z \in \mathbb{R}^m$, the softmax function, $s : \mathbb{R}^m \to (0,1)^m$, is defined as

$$s(z)_i = \frac{e^{z_i}}{\sum_{i=1}^m e^{z_i}}.$$

Let $y$ be a binary indicator (0 or 1) of the class label, and $c$ is the correct classification of the observation, $o$. Denote $p_{o,c}$ as the predicted probability of observation $o$ that belongs to class $c$. Then the negative log-loss function is defined as

$$\mathcal{L}_{LL}(y,p) = -\sum_{c=1}^m y_{o,c} \log(p_{o,c}).$$

**What loss functions and tasks do we cover?**  In general, for classification problems such as image classification by CNN, movie review prediction by RNN, and many more, the output layer is configured with a softmax function for prediction, and the negative log function is used as the loss function to train the DNN model. For binary classification, this loss is known as binary cross-entropy; for multi-class classification, it is called categorical cross-entropy. MSE is a consequence of Theorem A.1 with some modifications. Therefore, our analyses cover almost all the existing loss functions used for DNN training. We refrain from using some rarely used loss functions, e.g., sparse categorical cross-entropy.

## A.2  $\ell_2$ REGRESSION TASK

Next, we will prove Theorem A.1, for $\ell_2$-regression task. For Theorem A.1, the prediction of the DNN model does not involve any nonlinearity. Throughout Sections A.2 and A.3, $\mathbb{E}_p \cdot | \tilde{X}$ denotes expectation conditioned on the randomness in $R_p$ given $\tilde{X}$, and $\mathbb{E}_Z \cdot$ denotes expectation taken on the randomness in $\tilde{X}$.

**Theorem A.1.**  *With the notations above, we have* (i) $\mathbb{E}\|MX - MR_p(\tilde{X})\|_2^2 \leq \mathbb{E}\|MX - M\tilde{X}\|_2^2$ *if and only if* $p\|M \odot \bar{X}\|_F^2 + (1-p)\|MX\|_2^2 \leq \sigma^2\|M\|_F^2$, *where* $\bar{X} \in \mathbb{R}^{m \times n}$ *is a matrix obtained by stacking* $X^\top \in \mathbb{R}^{1 \times n}$ *in each row, and* $\odot$ *denotes the elementwise product.* (ii) *Let* $\alpha > 1$. $\mathbb{E}\|MX - MS_\alpha(\tilde{X})\|_2^2 \leq \mathbb{E}\|MX - M\tilde{X}\|_2^2$ *if and only if* $\frac{\|MX\|_2^2}{\|M\|_F^2} \leq \left(\frac{\alpha+1}{\alpha-1}\right)\sigma^2$.

*Remark* A.2.  Theorem A.1 considers the most commonly used mean square error (MSE), $\mathbb{E}\|MX - MR_p(\tilde{X})\|_2^2$ and $\mathbb{E}\|MX - MS_\alpha(\tilde{X})\|_2^2$, respectively, to compare against $\mathbb{E}\|MX - M\tilde{X}\|_2^2$. We use the MSE because it has nice mathematical properties; one can use other loss functions. This MSE is agnostic of the nature of the loss function used in DNN training.

*Remark* A.3. Since the expected MSE can be decomposed into bias and variance, by showing the relation between the expected MSEs as in Theorem A.1, the bias-variance trade-off between different processes can be explained.

**Proof of Theorem A.1.**

*Proof.* (*i*) We are required to show, $\mathbb{E}\|MX - M\tilde{X}\|_2^2 - \mathbb{E}\|MX - MR_p(\tilde{X})\|_2^2 \geq 0$. There are two types of randomness involved—one is due to randomness in $R_p$, and the second is due to the randomness in $\tilde{X}$. First, we start by writing

$$
\|MX - M\tilde{X}\|_2^2
$$

$$
= \|MX\|_2^2 - 2\langle MX, M\tilde{X}\rangle + \sum_{i=1}^m \sum_{j=1}^n m_{ij}^2 \tilde{x}_j^2 + 2\sum_{i=1}^m \sum_{1\leq j<k\leq n} m_{ij}m_{ik}\tilde{x}_j\tilde{x}_k, \tag{5}
$$

which after taking expectation becomes

$$
E[\|MX - M\tilde{X}\|_2^2]
$$

$$
= -\|MX\|_2^2 + \sum_{i=1}^m \sum_{j=1}^n m_{ij}^2(x_j^2 + \sigma^2) + 2\sum_{i=1}^m \sum_{1\leq j<k\leq n} m_{ij}m_{ik}x_jx_k. \tag{6}
$$

Next, we have as in (5)

$$
\|MX - MR_p(\tilde{X})\|_2^2
$$

$$
= \|MX\|_2^2 - 2\langle MX, MR_p(\tilde{X})\rangle + \sum_{i=1}^m \sum_{j=1}^n m_{ij}^2(R_p(\tilde{x}_j))^2 + 2\sum_{i=1}^m \sum_{1\leq j<k\leq n} m_{ij}m_{ik}R_p(\tilde{x}_j)R_p(\tilde{x}_k),
$$

$$
\tag{7}
$$

which after taking expectation conditioned on the randomness in $R_p$ given $\tilde{X}$ becomes

$$
E_p[\|MX - MR_p(\tilde{X})\|_2^2|\tilde{X}]
$$

$$
= \|MX\|_2^2 - 2p\langle MX, M\tilde{X}\rangle + p\sum_{i=1}^m \sum_{j=1}^n m_{ij}^2\tilde{x}_j^2 + 2p^2\sum_{i=1}^m \sum_{1\leq j<k\leq n} m_{ij}m_{ik}\tilde{x}_j\tilde{x}_k. \tag{8}
$$

Finally, taking the expectation on the randomness in $\tilde{X}$ we obtain

$$
\mathbb{E}_Z E_p[\|MX - MR_p(\tilde{X})\|_2^2|\tilde{X}]]
$$

$$
= (1-2p)\|MX\|_2^2 + p\sum_{i=1}^m \sum_{j=1}^n m_{ij}^2(x_j^2 + \sigma^2) + 2p^2\sum_{i=1}^m \sum_{1\leq j<k\leq n} m_{ij}m_{ik}x_jx_k. \tag{9}
$$

In view of equation 6 and equation 9, we have

$$
E[\|MX - M\tilde{X}\|_2^2] - E[\|MX - MR_p(\tilde{X})\|_2^2]
$$

$$
= (2p-2)\|MX\|^2 + (1-p)\sum_{i=1}^m \sum_{j=1}^n m_{ij}^2(x_j^2 + \sigma^2) + 2(1-p^2)\sum_{i=1}^m \sum_{1\leq j<k\leq n} m_{ij}m_{ik}x_jx_k
$$

$$
= (2p-2)\left[\sum_{i=1}^m \sum_{j=1}^n m_{ij}^2 x_j^2 + 2\sum_{i=1}^m \sum_{1\leq j<k\leq n} m_{ij}m_{ik}x_jx_k\right] + (1-p)\sum_{i=1}^m \sum_{j=1}^n m_{ij}^2(x_j^2 + \sigma^2)
$$

$$
+ 2(1-p^2)\sum_{i=1}^m \sum_{1\leq j<k\leq n} m_{ij}m_{ik}x_jx_k
$$

$$
= (1-p)\sum_{i=1}^m \sum_{j=1}^n m_{ij}^2(\sigma^2 - x_j^2) - 2(1-p)^2\sum_{i=1}^m \sum_{1\leq j<k\leq n} m_{ij}m_{ik}x_jx_k. \tag{10}
$$

Therefore,

$$\mathbb{E}\|MX - M\tilde{X}\|_2^2 - \mathbb{E}\|MX - MR_p(\tilde{X})\|_2^2 \geq 0$$

if and only if the expression in equation 10 is non-negative, that is,

$$\sum_{i=1}^m \sum_{j=1}^n m_{ij}^2 \sigma^2 \geq \sum_{i=1}^m \sum_{j=1}^n m_{ij}^2 x_j^2 + 2(1-p)\sum_{i=1}^m \sum_{1 \leq j < k \leq n} m_{ij}m_{ik}x_j x_k.$$

The left hand side of the above expression is $\sigma^2\|M\|_F^2$ (which is lower bounded by $n\sigma^2\sigma_{\min}^2(M)$, where $\sigma_{\min}(M)$ is the smallest singular value of $M$). For the right-hand side, we have

$$\sum_{i=1}^m \sum_{j=1}^n m_{ij}^2 x_j^2 + 2(1-p)\sum_{i=1}^m \sum_{1 \leq j < k \leq n} m_{ij}m_{ik}x_j x_k$$

$$= p\sum_{i=1}^m \sum_{j=1}^n m_{ij}^2 x_j^2 + (1-p)\sum_{i=1}^m \sum_{j=1}^n m_{ij}^2 x_j^2 + 2(1-p)\sum_{i=1}^m \sum_{1 \leq j < k \leq n} m_{ij}m_{ik}x_j x_k$$

$$= p\sum_{i=1}^m \sum_{j=1}^n m_{ij}^2 x_j^2 + (1-p)\|MX\|_2^2$$

$$= p\|M \odot \bar{X}\|_F^2 + (1-p)\|MX\|_2^2, \tag{11}$$

where $\bar{X} = \begin{pmatrix} x_1 & x_2 & x_3 & \cdots & x_n \\ x_1 & x_2 & x_3 & \cdots & x_n \\ \cdots & \cdots & \cdots & \cdots & \cdots \\ x_1 & x_2 & x_3 & \cdots & x_n \end{pmatrix} \in \mathbb{R}^{m \times n}$. Therefore,

$$\mathbb{E}\|MX - M\tilde{X}\|_2^2 - \mathbb{E}\|MX - MR_p(\tilde{X})\|_2^2 \geq 0$$

if and only if

$$\sigma^2\|M\|_F^2 \geq p\|M \odot \bar{X}\|_F^2 + (1-p)\|MX\|_2^2.$$

Hence the result.

(*ii*) We are required to show, $\mathbb{E}\|MX - M\tilde{X}\|_2^2 - \mathbb{E}\|MX - MS_\alpha(\tilde{X})\|_2^2 \geq 0$. Note that, the only randomness involved in this case is due to the randomness in $\tilde{X}$. First, we start by expanding

$$\|MX - MS_\alpha(\tilde{X})\|_2^2$$

$$= \|MX\|_2^2 - \frac{2}{\alpha}\langle MX, M\tilde{X}\rangle + \sum_{i=1}^m \sum_{j=1}^n m_{ij}^2 \frac{\tilde{x}_j^2}{\alpha^2} + 2\sum_{i=1}^m \sum_{1 \leq j < k \leq n} m_{ij}m_{ik}\frac{\tilde{x}_j\tilde{x}_k}{\alpha^2}, \tag{12}$$

which after taking expectation gives

$$E[\|MX - MS_\alpha(\tilde{X})\|_2^2]$$

$$= (1 - \frac{2}{\alpha})\|MX\|_2^2 + \sum_{i=1}^m \sum_{j=1}^n m_{ij}^2 \frac{(x_j^2 + \sigma^2)}{\alpha^2} + 2\sum_{i=1}^m \sum_{1 \leq j < k \leq n} m_{ij}m_{ik}\frac{x_j x_k}{\alpha^2}$$

$$= (1 - \frac{1}{\alpha})^2\|MX\|_2^2 + \frac{\sigma^2}{\alpha^2}\|M\|_F^2. \tag{13}$$

In view of equation 6 and equation 13, we have

$$E[\|MX - M\tilde{X}\|_2^2] - E[\|MX - MS_\alpha(\tilde{X})\|_2^2]$$

$$= (1 - \frac{1}{\alpha^2})\sigma^2\|M\|_F^2 - (1 - \frac{1}{\alpha})^2\|MX\|_2^2.$$

Therefore,

$$\mathbb{E}\|MX - M\tilde{X}\|_2^2 - \mathbb{E}\|MX - MS_\alpha(\tilde{X})\|_2^2 \geq 0$$

if and only if $(1 + \frac{1}{\alpha})\sigma^2\|M\|_F^2 - (1 - \frac{1}{\alpha})\|MX\|_2^2 \geq 0$. This completes our proof. □

## A.3 NONLINEAR LOSS FUNCTION FOR CLASSIFICATION TASK

Now, we will prove the results for the nonlinear loss function as given in Section 3.2. First, we quote the following Lemma about the moment generating function of a random variable, without proof. The readers can find the proof of Lemma A.4 in any standard graduate statistics textbook.

**Lemma A.4.** *Let $Z$ be a random variable, $Z \sim N(\mu, \sigma^2)$. Then the moment generating function, $\Phi_Z(\cdot)$ is given by $\Phi_Z(t) = \mathbb{E}e^{tZ} = e^{\mu t + \frac{\sigma^2 t^2}{2}}$.*

Calculating $\mathbb{E}\mathcal{L}_{LL}(y^\star, s(MR_p(\tilde{X})))$ requires some auxiliary results on the $\mathbb{E}\sum_{i=1}^{m} e^{(MR_p(\tilde{X}))_i}$ and $\mathbb{E}\left(\sum_{i=1}^{m} e^{(MR_p(\tilde{X}))_i}\right)^2$. The following Lemma A.5 gives the details, which are necessary for calculating the expectation and the variance of $\sum_{i=1}^{m} e^{(MR_p(\tilde{X}))_i}$.

**Lemma A.5.** *We have, (i) $\mathbb{E}\sum_{i=1}^{m} e^{(MR_p(\tilde{X}))_i} = \sum_{i=1}^{m} \prod_{k=1}^{n} \left( pe^{m_{ik}x_k + \frac{m_{ik}^2 \sigma^2}{2}} + (1-p) \right)$; and*

*(ii) $\mathbb{E}\left(\sum_{i=1}^{m} e^{(MR_p(\tilde{X}))_i}\right)^2 = \sum_{i,j} \left[ \prod_{k=1}^{n} \left( pe^{(m_{ik}+m_{jk})x_k + \frac{(m_{ik}+m_{jk})^2 \sigma^2}{2}} + (1-p) \right) \right].$*

*Proof.* (*i*) We have

$$\sum_{i=1}^{m} e^{(MR_p(\tilde{X}))_i} = \sum_{i=1}^{m} e^{\sum_{k=1}^{n} m_{ik} R_p(\tilde{x}_k)}, \tag{14}$$

where $\tilde{x}_k$ be the $k^{\text{th}}$ element of the vector $\tilde{X}$. Note that, each $m_{ik} R_p(\tilde{x}_k)$ is independent (based on the definition of the random masking operator), and after taking expectation on the above expression with respect to the randomness in $R_p$, we have

$$\mathbb{E}_p \sum_{i=1}^{m} e^{(MR_p(\tilde{X}))_i} = \sum_{i=1}^{m} \prod_{k=1}^{n} \mathbb{E}_p e^{m_{ik} R_p(\tilde{x}_k)}. \tag{15}$$

For $p \in (0, 1]$, equation 15 becomes

$$\mathbb{E}_p \sum_{i=1}^{m} e^{(MR_p(\tilde{X}))_i} = \sum_{i=1}^{m} \prod_{k=1}^{n} \mathbb{E}_p e^{m_{ik} R_p(\tilde{x}_k)} = \sum_{i=1}^{m} \prod_{k=1}^{n} \left( pe^{m_{ik}\tilde{x}_k} + (1-p)e^{m_{ik}\cdot 0} \right), \tag{16}$$

which further taking expectation on the randomenss in $\tilde{X}$ reduces to

$$\mathbb{E}_Z \mathbb{E}_p \sum_{i=1}^{m} e^{(MR_p(\tilde{X}))_i} = \sum_{i=1}^{m} \prod_{k=1}^{n} \left( \mathbb{E}_Z pe^{m_{ik}\tilde{x}_k} + (1-p) \right)$$

$$\overset{\text{Lemma } A.4}{=} \sum_{i=1}^{m} \prod_{k=1}^{n} \left( pe^{m_{ik}x_k + \frac{m_{ik}^2 \sigma^2}{2}} + (1-p) \right). \tag{17}$$

After taking total expectation on equation 17 and by using the tower property of expectation, we obtain the result.

(*ii*) We have

$$\left( \sum_{i=1}^{m} e^{(MR_p(\tilde{X}))_i} \right)^2 = \sum_{i,j} e^{\sum_{k=1}^{n} (m_{ik}+m_{jk}) R_p(\tilde{x}_k)}. \tag{18}$$

Proceeding similarly as above, first, taking expectation on the above expression with respect to the randomness in $R_p$ and then taking expectation with respect to the randomness in $\tilde{X}$, we have

$$
\begin{aligned}
\mathbb{E}_Z \mathbb{E}_p \left( \sum_{i=1}^m e^{(MR_p(\tilde{X}))_i} \right)^2 &= \mathbb{E}_Z \mathbb{E}_p \sum_{i,j} \left( e^{\sum_{k=1}^n (m_{ik}+m_{jk}) R_p(\tilde{x}_k)} \right) \\
&= \mathbb{E}_Z \sum_{i,j} \prod_{k=1}^n \mathbb{E}_p e^{(m_{ik}+m_{jk}) R_p(\tilde{x}_k)} \\
&= \mathbb{E}_Z \sum_{i,j} \prod_{k=1}^m \prod_{k=1}^n \left( p e^{(m_{ik}+m_{jk})\tilde{x}_k} + (1-p) \right) \\
&\overset{\text{Lemma A.4}}{=} \sum_{i,j} \prod_{k=1}^m \prod_{k=1}^n \left( p e^{(m_{ik}+m_{jk})x_k + \frac{(m_{ik}+m_{jk})^2 \sigma^2}{2}} + (1-p) \right).
\end{aligned}
$$
(19)

After taking total expectation on equation 19 and by using the tower property of expectation, we obtain the result.

$\square$

*Remark* A.6. Setting $p = 1$, in the loss function, we find the expected loss value, $\mathbb{E}\mathcal{L}_{LL}(y^\star, s(M\tilde{X}))$ due to noise injection (without random masking). Additionally, for $p = 1$, in Lemma A.5, we recover $\mathbb{E}\sum_{i=1}^m e^{(M\tilde{X})_i} = \sum_{i=1}^m \prod_{k=1}^n (e^{m_{ik}x_k + \frac{m_{ik}^2 \sigma^2}{2}})$.

The following Lemma[1], is the next intermediate result and instrumental in proving our main result as it approximates the expected logarithmic term in the log loss. In Lemma A.7, we approximate $\mathbb{E}[\log(x)]$ by using Taylor's Theorem.

**Lemma A.7.** *(Khuri, 2003, p. 117) Let $x$ be a positive random variable. Then $\mathbb{E}[\log(x)] = \log[\mathbb{E}(x)] - \frac{\mathrm{Var}(x)}{2(\mathbb{E}(x))^2} + higher\ order\ terms$, where $\mathrm{Var}(x) = \mathbb{E}(x^2) - (\mathbb{E}(x))^2$.*

*Remark* A.8. We assume that $x$ have small higher order moments, $m_p = \mathbb{E}|x - \mathbb{E}(x)|^p$, for $p = 2, 3, \cdots$.

Note that, setting $x = \sum_{i=1}^m e^{(MR_p(\tilde{X}))_i}$ in Lemma A.7 is the first step to quantify the expected loss value of SplitNN with random masking, $\mathbb{E}\mathcal{L}_{LL}(y^\star, s(MR_p(\tilde{X}))$.

### A.3.1 PROOF OF THEOREM 3.1

To prove Theorem 3.1, recall that $\mathcal{F}(p) := \mathbb{E}\mathcal{L}_{LL}(y^\star, s(MR_p(\tilde{X})))$, and consequently, $\mathcal{F}(1) = \mathbb{E}\mathcal{L}_{LL}(y^\star, s(M\tilde{X}))$. By using Lemma A.7 and assuming the higher order terms are negligible, we write

$$
\mathcal{F}(p) = \underbrace{-p(MX)_{i^*}}_{:=\mathcal{B}(p)} + \underbrace{\log \left( \mathbb{E} \sum_{i=1}^m e^{(MR_p(\tilde{X}))_i} \right)}_{:=\mathcal{C}(p)} - \underbrace{\frac{\mathrm{Var}\left( \sum_{i=1}^m e^{(MR_p(\tilde{X}))_i} \right)}{2 \left( \mathbb{E}\sum_{i=1}^m e^{(MR_p(\tilde{X}))_i} \right)^2}}_{:=\mathcal{D}(p)}.
$$
(20)

Differentiating equation 20 with respect to $p$ gives us:

$$
\mathcal{F}'(p) = \mathcal{B}'(p) + \mathcal{C}'(p) - \mathcal{D}'(p).
$$

Note that,

$$
\mathcal{B}'(p) = -(MX)_{i^*}, \mathcal{C}'(p) = \frac{\frac{d}{dp} \left( \mathbb{E}\sum_{i=1}^m e^{(MR_p(\tilde{X}))_i} \right)}{\mathbb{E}\sum_{i=1}^m e^{(MR_p(\tilde{X}))_i}},
$$

---

[1] See similar expression in Teh et al. (2006) with a restrictive assumption; assumption in Lemma A.7 is more general.

but the derivative of $\mathcal{D}(p)$ becomes very messy. So, we take the following indirect route: we first show that (i) $\mathcal{B}'(1) + \mathcal{C}'(1) \geq 0$ and hence $\mathcal{B}(p) + \mathcal{C}(p)$ are increasing in a neighborhood to the left of $p = 1$; then we verify that (ii) $-\mathcal{D}(p) \leq -\mathcal{D}(1)$. We see that once we accomplish (i) and (ii), we will have $\mathcal{F}(p) \leq \mathcal{F}(1)$, which completes the proof of Theorem 3.1.

**Proof of (i).**

By Lemme A.5 and a straightforward computation, we have

$$
\frac{d}{dp}\left(\mathbb{E}\sum_{i=1}^{m} e^{(MR_p(\tilde{X}))_i}\right)\Bigg|_{p=1} = \sum_{i=1}^{m}\sum_{r=1}^{n}\left[(e^{m_{ir}x_r + \frac{m_{ir}^2\sigma^2}{2}} - 1)\prod_{k\neq r, k=1}^{n} e^{m_{ik}x_k + \frac{m_{ik}^2\sigma^2}{2}}\right].
$$

So,

$$
\mathcal{B}'(1) + \mathcal{C}'(1) = -(MX)_i + \frac{\sum_{i=1}^{m}\sum_{r=1}^{n}\left[(e^{m_{ir}x_r + \frac{m_{ir}^2\sigma^2}{2}} - 1)\prod_{k\neq r, k=1}^{n}(e^{m_{ik}x_k + \frac{m_{ik}^2\sigma^2}{2}})\right]}{\sum_{i=1}^{m}\left(\prod_{k=1}^{n}(e^{m_{ik}x_k + \frac{m_{ik}^2\sigma^2}{2}})\right)},
$$

which is bigger than or equal to 0 if and only if

$$
\frac{\sum_{i=1}^{m}\sum_{r=1}^{n}\left[(e^{m_{ir}x_r + \frac{m_{ir}^2\sigma^2}{2}} - 1)\prod_{k\neq r, k=1}^{n}(e^{m_{ik}x_k + \frac{m_{ik}^2\sigma^2}{2}})\right]}{\sum_{i=1}^{m}\left(\prod_{k=1}^{n}(e^{m_{ik}x_k + \frac{m_{ik}^2\sigma^2}{2}})\right)} \geq (MX)_{i^*}, \tag{21}
$$

or equivalently,

$$
\sum_{i=1}^{m}\sum_{r=1}^{n}\left[(e^{m_{ir}x_r + \frac{m_{ir}^2\sigma^2}{2}} - 1)\prod_{k\neq r, k=1}^{n}(e^{m_{ik}x_k + \frac{m_{ik}^2\sigma^2}{2}})\right] \geq (MX)_{i^*}\sum_{i=1}^{m}\left(\prod_{k=1}^{n}(e^{m_{ik}x_k + \frac{m_{ik}^2\sigma^2}{2}})\right).
$$

Let $f(\sigma^2)$ denote the difference of the two sides, we have $f(\sigma^2) :=$

$$
\sum_{i=1}^{m}\left[\sum_{r=1}^{n}(e^{m_{ir}x_r + \frac{m_{ir}^2\sigma^2}{2}} - 1)\prod_{k\neq r, k=1}^{n}(e^{m_{ik}x_k + \frac{m_{ik}^2\sigma^2}{2}})\right] - (MX)_{i^*}\sum_{i=1}^{m}\prod_{k=1}^{n}(e^{m_{ik}x_k + \frac{m_{ik}^2\sigma^2}{2}})
$$

$$
= \sum_{i=1}^{m}\left(\prod_{k=1}^{n} e^{m_{ik}x_k + \frac{m_{ik}^2\sigma^2}{2}}\right)\left[-(MX)_{i^*} + \sum_{r=1}^{n}(1 - e^{-m_{ir}x_r - \frac{m_{ir}^2\sigma^2}{2}})\right]
$$

$$
= \sum_{i=1}^{m}\left(\prod_{k=1}^{n} e^{m_{ik}x_k + \frac{m_{ik}^2\sigma^2}{2}}\right)\left[\sum_{r=1}^{n}(1 - m_{i^*r}x_r - e^{-m_{ir}x_r - \frac{m_{ir}^2\sigma^2}{2}})\right].
$$

Now, note that $\sum_{r=1}^{n}(1 - m_{i^*r}x_r - e^{-m_{ir}x_r - \frac{m_{ir}^2\sigma^2}{2}})$ is an increasing function of $\sigma^2$ and as $\sigma^2 \to +\infty$, it approaches $\sum_{r=1}^{n}(1 - m_{i^*r}x_r) = n - \sum_{r=1}^{n} m_{i^*r}x_r$, which is positive by assumption. Thus, when $\sigma^2$ is large enough, $f(\sigma^2) > 0$. This verifies (21) and hence (i).

**Proof of (ii).**

We need to verify that $\mathcal{D}(p) \geq \mathcal{D}(1)$, that is, by the definition of $\mathcal{D}(p)$,

$$
\frac{\mathrm{Var}\left(\sum_{i=1}^{m} e^{(MR_p(\tilde{X}))_i}\right)}{2\left(\mathbb{E}\sum_{i=1}^{m} e^{(MR_p(\tilde{X}))_i}\right)^2} \geq \frac{\mathrm{Var}\left(\sum_{i=1}^{m} e^{(MR_p(\tilde{X}))_i}\right)}{2\left(\mathbb{E}\sum_{i=1}^{m} e^{(MR_p(\tilde{X}))_i}\right)^2}\Bigg|_{p=1}.
$$

By the formula $\mathrm{Var}(x) = \mathbb{E}x^2 - (\mathbb{E}x)^2$, it suffices to verify

$$\frac{\mathbb{E}\left(\sum_{i=1}^m e^{(MR_p(\tilde{X}))_i}\right)^2 - \left(\mathbb{E}\sum_{i=1}^m e^{(MR_p(\tilde{X}))_i}\right)^2}{2\left(\mathbb{E}\left(\sum_{i=1}^m e^{(MR_p(\tilde{X}))_i}\right)\right)^2}$$

$$\geq \left. \frac{\mathbb{E}\left(\sum_{i=1}^m e^{(MR_p(\tilde{X}))_i}\right)^2 - \left(\mathbb{E}\sum_{i=1}^m e^{(MR_p(\tilde{X}))_i}\right)^2}{2\left(\mathbb{E}\left(\sum_{i=1}^m e^{(MR_p(\tilde{X}))_i}\right)\right)^2}\right|_{p=1}.$$

which can be simplified to

$$\frac{\mathbb{E}\left(\sum_{i=1}^m e^{(MR_p(\tilde{X}))_i}\right)^2}{\left(\mathbb{E}\left(\sum_{i=1}^m e^{(MR_p(\tilde{X}))_i}\right)\right)^2} \geq \left. \frac{\mathbb{E}\left(\sum_{i=1}^m e^{(MR_p(\tilde{X}))_i}\right)^2}{\left(\mathbb{E}\left(\sum_{i=1}^m e^{(MR_p(\tilde{X}))_i}\right)\right)^2}\right|_{p=1}.$$

Using Lemma A.5, the above is the same as

$$\frac{\sum_{i,j}^m \prod_{k=1}^n \left[pe^{(m_{ik}+m_{jk})x_k+\frac{(m_{ik}+m_{jk})^2\sigma^2}{2}} + (1-p)\right]}{\left[\sum_{i=1}^m \prod_{k=1}^n \left(pe^{m_{ik}x_k+\frac{m_{ik}^2\sigma^2}{2}} + (1-p)\right)\right]^2} \geq \frac{\sum_{i,j}^m \prod_{k=1}^n e^{(m_{ik}+m_{jk})x_k+\frac{(m_{ik}+m_{jk})^2\sigma^2}{2}}}{\left(\sum_{i=1}^m \prod_{k=1}^n e^{m_{ik}x_k+\frac{m_{ik}^2\sigma^2}{2}}\right)^2} \quad (22)$$

We now verify (22) for $\sigma^2$ large enough. Note that (22) is true if and only if the following function $g(p) \geq 0$ where

$$g(p) := \sum_{i,j}^m \prod_{k=1}^n \left[pe^{(m_{ik}+m_{jk})x_k+\frac{(m_{ik}+m_{jk})^2\sigma^2}{2}} + (1-p)\right] \left(\sum_{i=1}^m \prod_{k=1}^n e^{m_{ik}x_k+\frac{m_{ik}^2\sigma^2}{2}}\right)^2$$

$$- \sum_{i,j}^m \prod_{k=1}^n \left(\left(e^{(m_{ik}+m_{jk})x_k+\frac{(m_{ik}+m_{jk})^2\sigma^2}{2}}\right)\right) \left[\sum_{i=1}^m \prod_{k=1}^n \left(pe^{m_{ik}x_k+\frac{m_{ik}^2\sigma^2}{2}} + (1-p)\right)\right]^2.$$

Note that $g(1) = 0$. So, it suffices to show that $g(p)$ is non-increasing on a small neighborhood to the left-hand side of 1. Differentiate $g$ to get

$$g'(1) = \sum_{i,j}^m \prod_{k=1}^n \left(e^{(m_{ik}+m_{jk})x_k+\frac{(m_{ik}+m_{jk})^2\sigma^2}{2}}\right) \sum_{r=1}^n \left(1 - e^{-(m_{ir}+m_{jr})x_r-\frac{(m_{ir}+m_{jr})^2\sigma^2}{2}}\right) \times I^2$$

$$- J \times 2I \times \sum_{i=1}^m \prod_{k=1}^n \left(e^{m_{ik}x_k+\frac{m_{ik}^2\sigma^2}{2}}\right) \sum_{r=1}^n \left(1 - e^{-m_{ir}x_r-\frac{m_{ir}^2\sigma^2}{2}}\right).$$

where $I := \sum_{i=1}^m \prod_{k=1}^n e^{m_{ik}x_k+\frac{m_{ik}^2\sigma^2}{2}}$ and $J := \sum_{i,j}^m \prod_{k=1}^n e^{(m_{ik}+m_{jk})x_k+\frac{(m_{ik}+m_{jk})^2\sigma^2}{2}}$. We have

$$g'(1) = I\left\{\sum_{i,j}^m \prod_{k=1}^n \left(e^{(m_{ik}+m_{jk})x_k+\frac{(m_{ik}+m_{jk})^2\sigma^2}{2}}\right) \sum_{r=1}^n \left(1 - e^{-(m_{ir}+m_{jr})x_r-\frac{(m_{ir}+m_{jr})^2\sigma^2}{2}}\right) \times I \right.$$

$$\left. - J \times 2 \times \sum_{i=1}^m \prod_{k=1}^n e^{m_{ik}x_k+\frac{m_{ik}^2\sigma^2}{2}} \sum_{r=1}^n \left(1 - e^{-m_{ir}x_r-\frac{m_{ir}^2\sigma^2}{2}}\right)\right\}$$

$$= I\left\{\sum_{i,j}^m \prod_{k=1}^n e^{(m_{ik}+m_{jk})x_k+\frac{(m_{ik}+m_{jk})^2\sigma^2}{2}} \left(n - \sum_{r=1}^n e^{-(m_{ir}+m_{jr})x_r-\frac{(m_{ir}+m_{jr})^2\sigma^2}{2}}\right) \times I \right.$$

$$\left. - J \times 2 \times \sum_{i=1}^m \prod_{k=1}^n e^{m_{ik}x_k+\frac{m_{ik}^2\sigma^2}{2}} \left(n - \sum_{r=1}^n e^{-m_{ir}x_r-\frac{m_{ir}^2\sigma^2}{2}}\right)\right\}$$

$$
= I \left\{ n \times I \times J - \sum_{i,j,s} \prod_{k=1}^{m} e^{(m_{ik}+m_{jk}+m_{sk})x_k + \frac{((m_{ik}+m_{jk})^2+m_{sk}^2)\sigma^2}{2}} \sum_{r=1}^{n} e^{-(m_{ir}+m_{jr})x_r - \frac{(m_{ir}+m_{jr})^2\sigma^2}{2}} \right.
$$

$$
\left. -2n \times J \times I + 2 \sum_{i,j,s=1}^{m} \prod_{k=1}^{n} e^{(m_{ik}+m_{jk}+m_{sk})x_k + \frac{((m_{ik}+m_{jk})^2+m_{sk}^2)\sigma^2}{2}} \sum_{r=1}^{n} e^{-m_{sr}x_r - \frac{m_{sr}^2\sigma^2}{2}} \right\}
$$

$$
= I \left\{ -n \times I \times J - \sum_{i,j,s} \prod_{k=1}^{m} e^{(m_{ik}+m_{jk}+m_{sk})x_k + \frac{((m_{ik}+m_{jk})^2+m_{sk}^2)\sigma^2}{2}} \sum_{r=1}^{n} e^{-(m_{ir}+m_{jr})x_r - \frac{(m_{ir}+m_{jr})^2\sigma^2}{2}} \right.
$$

$$
\left. +2 \sum_{i,j,s=1}^{m} \prod_{k=1}^{n} e^{(m_{ik}+m_{jk}+m_{sk})x_k + \frac{((m_{ik}+m_{jk})^2+m_{sk}^2)\sigma^2}{2}} \sum_{r=1}^{n} e^{-m_{sr}x_r - \frac{m_{sr}^2\sigma^2}{2}} ) \right\}
$$

$$
= I \left\{ \sum_{i,j,s} \prod_{k=1}^{m} e^{(m_{ik}+m_{jk}+m_{sk})x_k + \frac{((m_{ik}+m_{jk})^2+m_{sk}^2)\sigma^2}{2}} \right.
$$

$$
\left. \times \sum_{r=1}^{n} \left( -1 - e^{-(m_{ir}+m_{jr})x_r - \frac{(m_{ir}+m_{jr})^2\sigma^2}{2}} + 2e^{-m_{sr}x_r - \frac{m_{sr}^2\sigma^2}{2}} \right) \right\}
$$

The last sum above clearly goes to $-n$ as $\sigma^2 \to +\infty$. Thus, for $\sigma^2$ large enough, we have $g'(1) < 0$ which implies $g(p) \geq g(1) = 0$ for $p$ close to 1 from the left hand side. This completes the proof.

### A.3.2 Sketch of Proof of Theorem 3.2

The proof follows a similar line of arguments in the proof of Theorem 2 and thus, we point out only the main differences. We need to verify $\mathbb{E}\mathcal{L}_{LL}(y^\star, s(MS_\alpha(\tilde{X}))) \leq \mathbb{E}\mathcal{L}_{LL}(y^\star, s(M\tilde{X}))$ for $\alpha$ close to and larger than 1. With $\lambda = 1/\alpha$, one can show (as in Lemma 2) that

$$
\mathbb{E}\sum_{i=1}^{m} e^{MS_\alpha(\tilde{X})} = \sum_{i=1}^{m} e^{\lambda \sum_{k=1}^{n} m_{ik}x_k + \lambda^2\sigma^2 \sum_{k=1}^{n} m_{ik}^2/2}
$$

and

$$
\mathbb{E}\left( \sum_{i=1}^{m} e^{MS_\alpha(\tilde{X})} \right)^2 = \sum_{i=1}^{m} \sum_{j=1}^{m} e^{\lambda \sum_{k=1}^{n} (m_{ik}+m_{jk})x_k + \lambda^2\sigma^2 \sum_{k=1}^{n} (m_{ik}+m_{jk})^2/2}.
$$

Next, one can establish similar steps of (20), (21), and (22) for the current case and eventually complete the proof.

### A.3.3 Concentration of the Errors

**Linear layer.** Both conditions in Theorem 1 are necessary and sufficient conditions and indicate implicit relations between the input, $X$, the denoising parameters, $p, \alpha$, the weight of the split layer, $W$, and the added noise magnitude $\alpha$. For the masking case, the coefficient of the linear term, $p$, which can be positive or negative depending on the noise scale $\sigma$, dominates MSE; see Figure 2 (b). However, for Theorem 1 (ii), from (11), we observe the MSE depends quadratically on the scaling factor $\frac{1}{\alpha}$, where $\alpha > 1$. Therefore, one can observe a quadratic relation between the scaling factor $\frac{1}{\alpha}$, and MSE in Figure 2(a).

**Nonlinear layer.** For nonlinear loss functions, this relation is more complicated to observe. For Theorem 2, the loss is $\mathcal{L}_{LL}(y^\star, s(MR_p(\tilde{X})))$, which is given in equation (2) in the main paper (also, see equation (18) in the Appendix). In this scope, we show how the bound on the error, $\mathbb{E}[\mathcal{L}_{LL}(y^\star, s(M\tilde{X})) - \mathcal{L}_{LL}(y^\star, s(MR_p(\tilde{X})))]$, depends on $\sigma^2$ and $(1-p)$. Based on equation (18), the first term will be $-(1-p)(MX)_{i^\star}$, which is linear in $(1-p)$. The next term we need to consider is $\log\left( \mathbb{E}[\sum_{i=1}^{m} e^{(MR_p(\tilde{X}))_i}] \right) - \log\left( \mathbb{E}[\sum_{i=1}^{m} e^{(MX)_i}] \right)$. By directly manipulating the expression in

(15) for $p$ and $p = 1$ we find this quantity approximately is

$$\sum_{i=1}^{m} e^{\sum_{k=1}^{n} m_{ik}x_k} \{(p^n - 1) + \sigma^2 p^n \frac{\sum_{k=1}^{n} m_{ik}^2}{2} + \text{Higher order terms}\} + O(1 - p).$$

When the noise is not too large, say, $\sigma^2 \frac{\sum_{k=1}^{n} m_{ik}^2}{2} < 1$, we can observe that the bound is approximately linear in the variance of noise, $\sigma^2$. One can obtain a similar observation for Theorem 3 as well.

### A.4 DIFFERENTIAL PRIVACY (DP)

We start by defining differential privacy.

**Definition A.9.** Dwork et al. (2014) A random mechanism, $\mathcal{K} : \mathcal{D} \to \mathbb{R}^m$ is $(\epsilon, \delta)$-differentially private if for all adjacent inputs, $D, D' \in \mathcal{D}$, with Hamming distance, $d(D, D') = 1$, and all possible output, $\mathcal{O} \in \mathbb{R}^m \in \mathcal{B}(\mathbb{R}^m)$ such that

$$P[\mathcal{K}(D) \in \mathcal{O}] \leq e^\epsilon P[\mathcal{K}(D') \in \mathcal{O}] + \delta,$$

where $P[\cdot]$ refers to the probability associated with the random mechanism $\mathcal{K}$ and $\mathcal{B}(\mathbb{R}^m)$ is Borel sets in $\mathbb{R}^m$.

We also need the general theorem that says, any differentially private mechanism, $\mathcal{K} : \mathcal{D} \to \mathcal{O}$ is further differentially private if it is transformed by an arbitrary postprocessing function, deterministic or random; see Theorem 3.4 in the main paper.

Finally, if perform multiple computations of the random mechanism, $\mathcal{K}$ on the same dataset, $D$, that is, we make $T$ such passes on $D$, then the privacy guarantee degrades. We quote the advanced composition theorem from Dwork et al. (2010) for such mechanisms.

**Theorem A.10.** *Dwork et al. (2010) Let $\mathcal{K} : D \to \underbrace{\mathcal{O} \times \mathcal{O} \cdots \mathcal{O}}_{T-\text{times}}$ be an $T$-fold adaptive composition of $(\epsilon, \delta)$-DP mechanisms. Then $\mathcal{K}$ is $(\epsilon', T\delta + \delta')$-DP for $\epsilon' = \epsilon\sqrt{2T \ln(\delta'^{-1})} + T\epsilon(e^\epsilon - 1)$, for all $\delta' > 0$.*

#### A.4.1 DP OF THE GRADIENT DURING BACKPROPAGATION

**Setup.** Let $\{(X_i, y_i^\star)\}_{i=1}^N$ be training data points. Let the $k^{\text{th}}$ layer of an $L$ layer DNN be $X_i^k$, let $\Phi$ be a differentiable activation function, and let $M_k^t$ be the weight matrix for the $k^{\text{th}}$ layer at iteration $t$. By this convention, $X_i = X_i^0$. At $t = 0$, we have

$$X_i^j = \Phi_j(M_j^0 X_i^{j-1}), \ \ y_i^j = M_j^0 X_i^{j-1}, \ \ j = 1, \cdots L. \tag{23}$$

For the noisy split neural network with post-processing, let the split happen at the $(l-1)^{\text{th}}$ layer (so, the cut layer is the $(l-1)^{\text{th}}$ layer). With the notations above, we have:

$$\begin{cases} X_i^j = \Phi_j(M_j^0 X_i^{j-1}), \ y_i^j = M_j^0 X_i^{j-1}, \ j = 1, \cdots l - 1, \\ \tilde{X}_i^{l-1} = X_i^{l-1} + \Delta, \tilde{X}_i^l = \Phi_l(M_l^0 h_D(\tilde{X}_i^l)), \tilde{y}_i^l = M_l^0 h_D(\tilde{X}_i^l), \\ \text{and } \tilde{X}_i^k = \Phi_k(M_k^0 \tilde{X}_i^{k-1}), \tilde{y}_i^k = M_k^0 \tilde{X}_i^k, k = l+1, \cdots L. \end{cases} \tag{24}$$

In our experiments, we use the regular backpropagation formula in our noisy split network training, which we formalize in the next Proposition.

**Proposition A.11.** *With the notations above, for a noisy split network with post-processing, during training, one can use the regular backpropagation algorithm by substituting: (i) $y_i^k = \tilde{y}_i^k, X_i^k = \tilde{X}_i^k$ for all subsequent $k = L - l + 1, \cdots, L$ and (ii) $y_i^r = y_i^r, X_i^r = X_i^r$ for all $r = 1, 2, \cdots, l - 1$, before the split, and $y_i^k = \tilde{y}_i^k$, for $k = L - l + 1, \cdots, L$ after the split.*

Define the sets, $S \subseteq \mathbb{R}^m$ and $S^c$ as follows:

$$\begin{cases} S := \{z : p(z) > e^\epsilon p(z + V(D) - V(D'))\}, \\ S^c := \{z : p(z) \leq e^\epsilon p(z + V(D) - V(D'))\}, \end{cases} \tag{25}$$

**Proposition A.12.** *Let $\epsilon, \delta > 0$, and let a random mechanism $\mathcal{K} : D \to \mathbb{R}^m$ be defined as in (3). Then for any two adjacent datasets, $D$ and $D'$, large enough, we have that the random variable $Z$ will satisfy $P[\omega : p(Z(\omega)) > e^\epsilon p(Z(\omega) + V(D) - V(D'))] < \delta$.*

The above Proposition can be verified as follows. Because, for two large enough datasets, $D$ and $D'$, we can make $\Delta := \|V(D) - V(D')\|$ small. Using Taylor expansion, we have

$$p(z) - e^\epsilon p(z + V(D) - V(D'))$$
$$= p(z) - e^\epsilon (p(z) + \nabla p(z)^\top (V(D) - V(D')) ) + o(\Delta^2)$$
$$\approx (1 - e^\epsilon)p(z) + O(\Delta) \leq 0.$$

**Theorem A.13.** *Let a random mechanism, $\mathcal{K} : D \to \mathbb{R}^m$ as defined in equation 3 obey Proposition A.12. Then for some $(\epsilon, \delta)$, $\mathcal{K}$ is $(\epsilon, \delta)$-DP.*

*Proof.* Note that, $P[\mathcal{K}(D) \in \mathcal{O}] = P[V(D) + z \in \mathcal{O}] = P[z \in \mathcal{O} - V(D)] \overset{\mathcal{O}' := \mathcal{O} - V(D)}{=} P[z \in \mathcal{O}']$, where $\mathcal{O}'$ is a shifted output set. We split the set $\mathcal{O}'$ into two disjoint sets, $\mathcal{O}' \cap S$ and $\mathcal{O}' \cap S^c$. Therefore,

$$
\begin{aligned}
P[\mathcal{K}(D) \in \mathcal{O}] = P[z \in \mathcal{O}'] \quad &= \quad P[z \in ((\mathcal{O}' \cap S) \cup (\mathcal{O}' \cap S^c))] \\
&= \quad P[z \in (\mathcal{O}' \cap S)] + P[z \in (\mathcal{O}' \cap S^c)] \\
&\overset{\mathcal{O}' \cap S \subseteq S}{\leq} \quad P[z \in S] + P[z \in (\mathcal{O}' \cap S^c)] \\
&\overset{\text{Proposition A.12}}{<} \quad \delta + P[z \in (\mathcal{O}' \cap S^c)].
\end{aligned}
\tag{26}
$$

We also have

$$
\begin{aligned}
P[z \in (\mathcal{O}' \cap S^c)] = \int_{z \in \mathcal{O}' \cap S^c} p(z) dz \quad &\overset{\text{By equation 25}}{\leq} \quad e^\epsilon \int_{z \in \mathcal{O}' \cap S^c} p(z + V(D) - V(D')) dz \\
&\overset{\mathcal{O}' \cap S^c \subseteq \mathcal{O}'}{\leq} \quad e^\epsilon \int_{z \in \mathcal{O}'} p(\underbrace{z + V(D) - V(D')}_{=u}) dz \\
&= \quad e^\epsilon \int_{u \in \mathcal{O} - V(D')} p(u) du \\
&= \quad e^\epsilon \int_{z + V(D') \in \mathcal{O}} p(z) dz \\
&= \quad e^\epsilon P[V(D') + z \in \mathcal{O}] \\
&= \quad e^\epsilon P[\mathcal{K}(D') \in \mathcal{O}].
\end{aligned}
\tag{27}
$$

Combining equation 26 and equation 27, we get the result. $\qquad\square$

Let $g_B^t(D)$ and $\tilde{g}_B^t(D)$ be two gradient vectors computed on the training dataset $D$ over a minibatch $B$ at iteration $t$ without and with noise injected. Based on Proposition A.11 we have:
$$
\begin{cases}
g_B^t(D) = g(y_B^L(D), y_B^{L-1}(D), y_B^{L-2}(D), \cdots, y_B^1(D)). \\
\tilde{g}_B^t(D) = g(\tilde{y}_B^L(D), \cdots, \tilde{y}_B^l(D), y_B^{l-1}(D), \cdots, y_B^1(D)).
\end{cases}
$$

Define $Z := \tilde{g}_B^t(D) - g_B^t(D)$ be a random vector. Hence,
$$\mathcal{K}_{BP}(D) := \tilde{g}_B^t(D) = g_B^t(D) + Z, \tag{28}$$

where $Z = G(\xi, \beta)$ is a random variable with probability density function $p(\tilde{z} \in \mathcal{O}) = p((\xi, \beta) \in G^{-1}(\mathcal{O}))$. In our case, we use masking and scaling as a postprocessing function, $h_D(\cdot)$ after the Gaussian noise injection. The random variable, $\xi \sim \mathcal{N}(0, \sigma I)$ is continuous. The mask, $R_p$ is a random matrix of 1 and 0 with identical and independently distributed entries, $(R_p)_{ij} \sim \text{Bernoulli}(p)$, a discrete distribution. For scaling, $S_\alpha$ is an elementwise scaling operator. Therefore, for masking, $\tilde{y}_i^l = M_l^0 R_p(\tilde{X}_i^l)$, and the entries of $R_p(\tilde{X}_i^l) = X_i^{l-1}(D) + \Delta$ or 0, based on $(R_p)_{ij} \sim \text{Bernoulli}(p)$. On the other hand, for scaling, $\tilde{y}_i^l = M_l^0 S_\alpha(\tilde{X}_i^l)$ and the entries of $S_\alpha(\tilde{X}_i^l) = \frac{1}{\alpha}(X_i^{l-1}(D) + \Delta)$. Therefore, based on Theorem A.13, this mechanism is also differentially private.

| Model & Dataset | Split Config. (denoted by ‖ ) | Layer Size |
|---|---|---|
| CNN on MNIST | 2×Conv2d - FC ‖ FC - Loss | 256 |
| | 2×Conv2d ‖ FC - FC - Loss | 12544 |
| MLP on IMDB | Embedding - FC ‖ FC - Loss | 16 |
| | Embedding ‖ FC - FC - Loss | 4096 |
| ResNet-20 on CIFAR10 | Conv2d - 3×ResBlock ‖ FC - Loss | 256 |
| ResNet-18 on CIFAR100 | Conv2d - 4×ResBlock ‖ FC - Loss | 512 |
| RNN on Names | Embedding - RNN ‖ FC - Loss | 256 |
| ResNet-50 on ImageNet1K | Conv2d - 4×ResBlock ‖ FC - Loss | 512 |
| ALBERT-base-v2 on Amazon Reviews | 12×Encoder ‖ FC - Loss | 768 |

Table 1: Model split configurations and split layer sizes.

Table 2: SplitNN setup and training hyper-parameters

| Model | Dataset | Optimizer | Batch size | Epoch | lr | Weight decay |
|---|---|---|---|---|---|---|
| CNN | MNIST | SGD | 64 | 4 | 0.1 | 0 |
| ResNet-20 | CIFAR-10 | SGD-M | 128 | 160 | 0.1 | 1e-4 |
| ResNet-18 | CIFAR-100 | SGD-M | 128 | 200 | 0.1 | 5e-4 |
| MLP | IMDB | Adam | 64 | 2 | 0.01 | 0 |
| LSTM | Names | SGD | 800 | 150 | 2 | 0 |
| ALBERT-base-v2 | Amazon Reviews | AdamW | 256 | 10 | 5e-5 | 1e-2 |
| ResNet-50 | ImageNet1K | SGD | 256 | 90 | 0.1 | 1e-4 |

# B    ADDENDUM TO THE NUMERICAL RESULTS

Due to limited space, we were unable to discuss many experimental details as well as many results in Section 4 of the main paper. We discuss them here in detail.

## B.1    LAPLACE MECHANISM

The Laplace Distribution (centered at 0) with scale $b$ is the distribution with probability density function:

$$\text{Lap}(x \mid b) = \frac{1}{2b} \exp\left(-\frac{|x|}{b}\right)$$

Here, we consider the Laplace mechanism to protect the input vector $X$. Simulation results (Figure 7) show that our denoising methods can also decrease the estimation error caused by Laplace mechanism during forward and backward pass. However, in the real split learning task (see Table 3), our denoising methods are less effective for large Laplacian noise $b = 0.7$, compared with $\sigma = 0.7$ in Gaussian mechanism. More detailed investigation is left for future work.

## B.2    SPLIT LEARNING RESULT ON CIFAR-100

We provide results on the CIFAR-100 dataset with ResNet-18 in Table 5. As shown in the table, both our denoising techniques achieve higher accuracy than simple learning rate tuning.

Table 3: Denoising performance by using Laplacian noise in split learning for MNIST classification task (same experiment setting as Figure 3a). We fine-tune hyperparameters $\lambda$ and $p$ for different Laplacian noise scales $b = 0.3, 0.5, 0.7$. Compared withthe Gaussian mechanism, split learning suffers more from Laplacian noise injection.

| $b$ | Best acc. (%) | $\lambda$=0.1 | $\lambda$=0.2 | $\lambda$=0.4 | $\lambda$=0.6 | $p$=0.1 | $p$=0.2 | $p$=0.4 | $p$=0.6 |
|---|---|---|---|---|---|---|---|---|---|
| 0 | 98.96 (±0.13) | - | - | - | - | - | - | - | - |
| 0.3 | 92.64 (±0.11) | 95.66 (±0.25) | 94.49 (±0.31) | 92.16 (±0.38) | 90.93 (±0.49) | 98.19 (±0.67) | 98.55 (±0.62) | 98.32 (±0.17) | 95.54 (±0.21) |
| 0.5 | 38.30 (±0.27) | 94.58 (±0.18) | 93.44 (±0.27) | 89.11 (±0.53) | 85.10 (±0.35) | 96.98 (±0.39) | 97.48 (±0.84) | 95.17 (±0.31) | 89.01 (±0.37) |
| 0.7 | 16.62 (±0.10) | 21.44 (±0.13) | 23.49 (±0.58) | 20.52 (±0.34) | 14.71 (±0.52) | 12.85 (±0.27) | 23.62 (±0.45) | 18.42 (±0.49) | 15.58 (±0.19) |

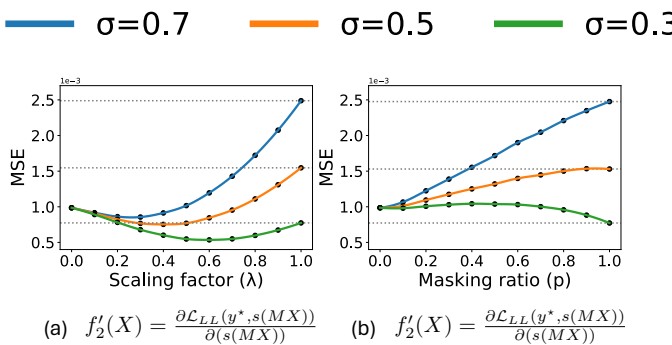

(a) $f_2'(X) = \frac{\partial \mathcal{L}_{LL}(y^\star, s(MX))}{\partial(s(MX))}$  (b) $f_2'(X) = \frac{\partial \mathcal{L}_{LL}(y^\star, s(MX))}{\partial(s(MX))}$

Figure 6: **Backward simulation.** Simulation of how scaling factor ($\lambda = \frac{1}{\alpha}$) and masking ratio ($p$) influence the estimation error (MSE) under different noise levels ($\sigma$) for linear ($f_1$) and nonlinear functions ($f_2, f_2'$). The backpropagation errors are essential during the training as we use them together with forward IRs to directly compute the gradients. By taking the derivative of the loss function w.r.t the IRs, we obtain the simulation of the MSEs for the backpropagation errors. Similar to the forward pass, scaling and masking, can lower the estimation error during the backward pass, especially when the noise level is relatively high.

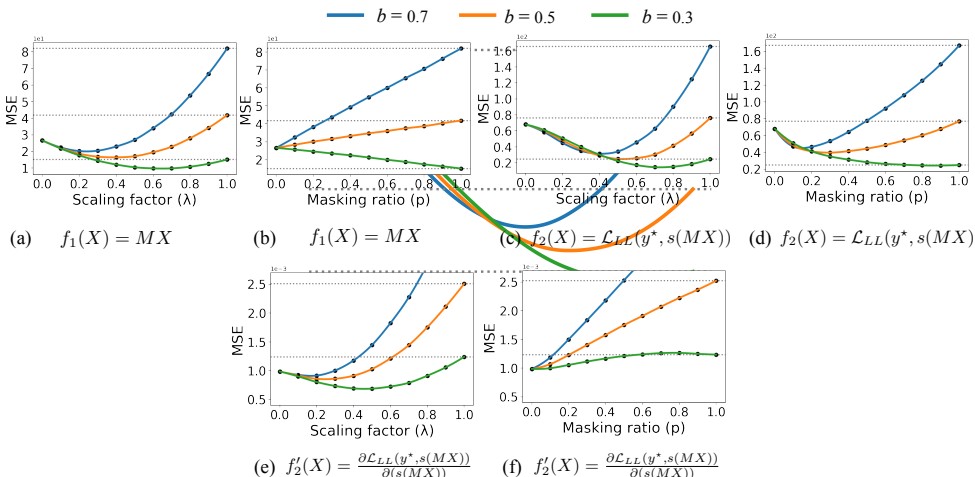

(a) $f_1(X) = MX$  (b) $f_1(X) = MX$  (c) $f_2(X) = \mathcal{L}_{LL}(y^\star, s(MX))$  (d) $f_2(X) = \mathcal{L}_{LL}(y^\star, s(MX))$

(e) $f_2'(X) = \frac{\partial \mathcal{L}_{LL}(y^\star, s(MX))}{\partial(s(MX))}$  (f) $f_2'(X) = \frac{\partial \mathcal{L}_{LL}(y^\star, s(MX))}{\partial(s(MX))}$

Figure 7: Simulation of how scaling factor ($\lambda = \frac{1}{\alpha}$) and masking ratio ($p$) influence the estimation error (MSE) under different Laplacian noise levels ($b$) for linear and nonlinear cases. Plots (a)–(d) are the linear layer and nonlinear layer during the forward pass, while (e)-(f) are the derivatives of the nonlinear layer during the backward pass.

### B.3 UNDERSTANDING THE SCALING OPERATION WHEN GRADIENT SCALE IMPACT IS ELIMINATED

To further understand the scaling operations, we run the same MNIST experiment with two Adam optimizers separately for the client and server. The results in Table 6 demonstrate that the efficacy of scaling operation still exists even when gradient scale impact is eliminated by Adam's updating rule. If we compare our scaling method against learning rate tuning, the accuracy gain should be from 92.80% to 98.14%.

### B.4 SPLIT LEARNING ON LARGE DATASETS

We use two large-scale datasets, Amazon Reviews McAuley & Leskovec (2013) for sentiment analysis and ImageNet1K Deng et al. (2009) for image classification tasks, and investigate how the proposed denoising strategies perform in large-scale, challenging datasets. For the experiments on the Amazon Reviews dataset, we use a pre-trained ALBERT-base-v2 Lan et al. (2020); for the ImageNet1K

| Learning rate | 0.001 | 0.005 | 0.01 | 0.05 | 0.1 |
|---|---|---|---|---|---|
| Top-1 Acc. (%) | 87.68 | **92.67** | 81.21 | diverge | diverge |
| Weight decay | 0.01 | 0.05 | 0.1 | 0.2 | 0.4 |
| Top-1 Acc. (%) | diverge | diverge | diverge | **87.15** | 77.78 |
| Dropout | 0.1 | 0.2 | 0.4 | 0.6 | 0.8 |
| Top-1 Acc. (%) | diverge | diverge | diverge | diverge | diverge |
| Masking | 0.1 | 0.2 | 0.4 | 0.6 | 0.8 |
| Top-1 Acc. (%) | 98.31 | **98.62** | diverge | diverge | diverge |
| Scaling* | 0.1 | 0.2 | 0.4 | 0.6 | 0.8 |
| Top-1 Acc. (%) | **98.10** | 96.06 | 90.25 | diverge | diverge |

Table 4: Comparison of tuning various hyper-parameters in noise injected SplitNN training at a fixed noise level ($\sigma = 0.7$) for MNIST classification. * means co-optimization with weight decay.

Table 5: Top-1 Accuracy (%) of split learning using ResNet-18 on CIFAR-100 dataset with noise injection level ($\sigma = 0.7$). The baseline achieves 75.60% Top-1 accuracy using SGD-Momentum (m=0.9) with an initial learning rate of 0.1, and weight decay $5e^{-4}$.

| Learning rate | 0.001 | 0.005 | 0.01 | 0.05 | 0.1 |
|---|---|---|---|---|---|
| Top-1 Acc. (%) | 53.11 | **68.93** | 61.54 | diverge | diverge |
| Masking (lr=0.1) | 0.1 | 0.2 | 0.4 | 0.6 | 0.8 |
| Top-1 Acc. (%) | 60.49 | **72.38** | 63.31 | 52.97 | diverge |
| Scaling (lr=0.1) | 0.1 | 0.2 | 0.4 | 0.6 | 0.8 |
| Top-1 Acc. (%) | **72.21** | 71.68 | 68.14 | 57.54 | diverge |

experiments, we use ResNet50; see Table 7 for the results. We provide the split configurations and the training hyperparameters for these experiments in Tables 1 and 2, respectively. As the ALBERT-base-v2 was a pre-trained model, we fine-tuned it on the Amazon Reviews dataset, and observed that noise injections or denoising strategies, such as masking and scaling, had an insignificant effect on the baseline performance. Noise injections do not degrade the performance, and denoising strategies do not improve them either. On the other hand, the performance of ResNet50 on ImageNet1K was heavily impacted by the noise injection. E.g., a noise injection of $\sigma = 0.5$ renders a test accuracy of 1.92, which is 97% lower than the baseline accuracy. When we apply the denoising strategies to the noise-injected IRs, at the same noise level $\sigma = 0.5$, the scaling strategy with $\lambda = 0.1$ and masking strategy with $p = 0.2$ recover performance comparable to the baseline.

## B.5 EXPERIMENTS ON SPLITTING THE NETWORK

We investigated the network behavior in the split setup and experimented with different layers to split the network. For this, we selected the configurations of MLP on the IMDB dataset and ResNet20 on the CIFAR10 dataset with the highest noise-level, $\sigma = 0.7$; see Table 8 for the results. We observe that splitting the networks at the initial layers causes a performance degradation compared to when the networks are split towards the end. We also observe that for MLP architectures, the scaling strategy is more effective for denoising, while the masking strategy is more effective for ResNet architectures.

Table 6: Top-1 Accuracy(%) of split learning on MNIST dataset with noise injection level ($\sigma = 0.7$). Use two Adam optimizers separately for the client and server. The Adam baseline (lr=1e-3) achieves 98.95% top-1 accuracy.

| Learning rate | 1e-6 | 1e-5 | 1e-4 | 1e-3 | 0.1 |
|---|---|---|---|---|---|
| Top-1 Acc. (%) | 83.96 | **92.80** | 90.13 | diverge | diverge |
| Scaling ratio (lr=1e-4) | 0.1 | 0.2 | 0.4 | 0.6 | 0.8 |
| Top-1 Acc. (%) | 97.19 | 97.94 | **98.14** | 96.03 | 91.75 |

Table 7: Performance of scaling ($\lambda$) and masking ($p$) at different noise levels ($\sigma$) for large-scale datasets, ImageNet and Amazon Reviews Full.

| Task | Noise level ($\sigma$) | Best acc. (%) | $\lambda$=0.1 | $p$=0.2 |
|---|---|---|---|---|
| | 0 | **74.85** | - | - |
| ResNet50-ImageNet | 0.3 | 31.18 | 74.26 | 73.79 |
| | 0.5 | 1.92 | 72.93 | 73.07 |
| | 0 | **65.19** | - | - |
| ALBERT-base-v2-Amazon Reviews | 0.3 | 65.15 | 65.12 | 65.15 |
| | 0.5 | 65.18 | 65.16 | 65.13 |

Table 8: Performance of scaling ($\lambda$) and masking ($p$) for different splits of the networks.

| Task | Splits | $\sigma$ | Best acc. (%) | $\lambda$=0.1 | $\lambda$=0.2 | $p$=0.1 | $p$=0.2 |
|---|---|---|---|---|---|---|---|
| | - | 0 | **85.53** | - | - | | |
| MLP-IMDB | Embedding - FC ǁ FC - Loss | 0.7 | 76.28 | 83.25 | 82.35 | 82.82 | 83.41 |
| | Embedding ǁ FC - FC - Loss | 0.7 | 69.08 | 73.61 | 72.91 | 64.95 | 66.12 |
| | - | 0 | **91.76** | - | - | | |
| ResNet20-CIFAR10 | Conv2d - 3×ResBlock ǁ FC - Loss | 0.7 | 80.69 | 86.69 | 87.16 | 89.07 | 89.48 |
| | Conv2d - 2×ResBlock ǁ 1×ResBlock - FC - Loss | 0.7 | 66.16 | 35.53 | 38.22 | 85.01 | 86.95 |

### B.6 COMPUTATIONAL OVERHEAD OF THE POSTPROCESSING FUNCTIONS

We measured the computational overhead introduced by the denoising techniques on both CPU and GPU. The results are shown in Table 9. The computational overhead of scaling and masking is negligible compared to the training computation.

Table 9: Run time profiling for one mini-batch training of CNN on MNIST dataset. GPU: NVIDIA A100-80GB GPU. CPU: Intel Xeon Platinum 8260 CPU @ 2.40GHz.

| Hardware | Baseline | Noise injection only | Noise injection w. masking | Noise injection w. scaling |
|---|---|---|---|---|
| GPU | 1.54 ms | 1.88 ms | 1.91 ms | 1.92 ms |
| CPU | 22.16 ms | 22.27 ms | 23.40 ms | 23.19 ms |

### B.7 FEATURE-SPACE HIJACKING ATTACK (FSHA) AND OUR POST-PROCESSING TECHNIQUES

**Threat model.** We assume that the attacker has no information on the architecture of the client's model and its weights. However, the attacker knows a public dataset that captures the same domain of the clients' training sets. For example, if the model is trained on face images, then the public dataset is composed of face images as well. This assumption is more realistic and less restrictive than the ones adopted in other works Vepakomma et al. (2019)Vepakomma et al. (2018b), where the attacker is assumed to have direct access to leaked pairs of intermediate results and private training data.

In the FSHA Pasquini et al. (2021) attack, the attacker (e.g. the server) can hijack the client's learning process and learn an inverse version of the client's model. During the inference, the attacker can recover the client's raw data by using the output of the client. In this work, we showed that the masking operator simultaneously improves the SplitNN training and in the meantime, decreases the efficacy of the FSHA attack. Our intuition is that the denoising effect depends on the specific application, which is a function applied to the noisy input, $X + \Delta$. If the function's goal is to identify each value of $X$, such as reconstructing an image in an FSHA attack, then denoising cannot help too much. However, if the goal is to get a more accurate estimation on some statistical metrics of $X$, such as the mean, norm of $X$, then it is possible to have an evident denoising improvement.

### B.8 DP BUDGET

We provide the privacy bounds for one single forward pass during the SplitNN training in Table 11. By using the general composition Theorem, Theorem A.10 in the Appendix, we can calculate the total privacy budget for the entire training process. Although Theorem A.10 gives a theoretical formalization, in practice, if we use Theorem A.10 directly, it will result in a large privacy bound, which may not be practical. As argued in the paper DP-SGD, Abadi et al. found that even for

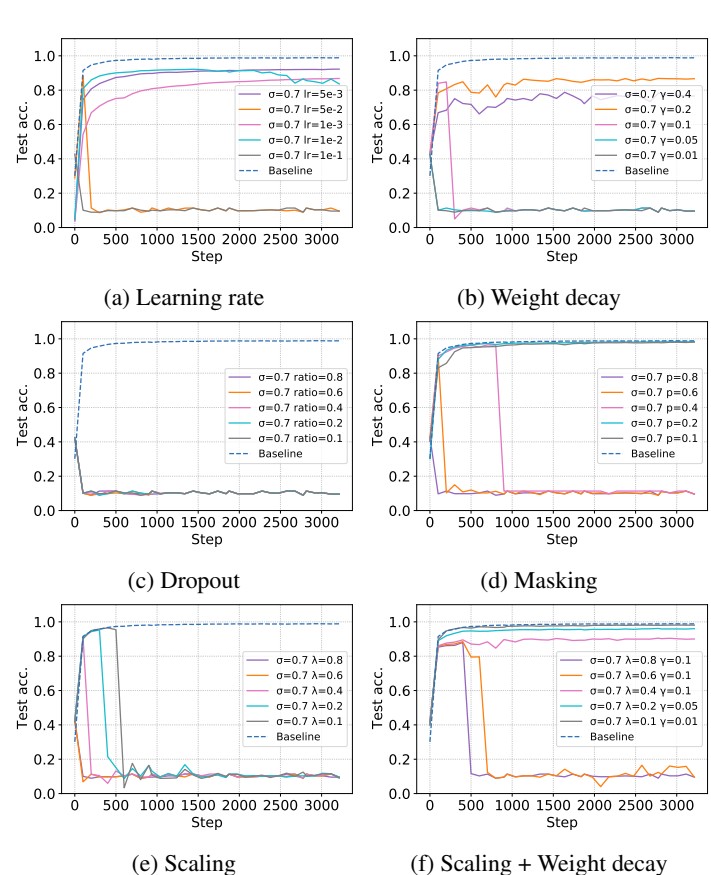

Figure 8: Comparison of tuning various hyper-parameters in noise-injected split learning at a fixed noise level ($\sigma = 0.7$) for MNIST classification task.

Table 10: Hyper-parameter tuning for scaling ($\lambda$) and masking ($p$) at different noise level ($\sigma$). Results are obtained by running the experiment for 3 times with different random seeds. We record the best test accuracy during the training instead of the final accuracy.

| Task | $\sigma$ | Best acc. (%) | $\lambda=0.1$ | $\lambda=0.2$ | $\lambda=0.4$ | $\lambda=0.6$ | $p=0.1$ | $p=0.2$ | $p=0.4$ | $p=0.6$ |
|---|---|---|---|---|---|---|---|---|---|---|
| CNN-MNIST | 0 | **98.96 (±0.13)** | - | - | - | - | - | - | - | - |
| | 0.3 | 98.46 (±0.07) | 97.63 (±0.27) | 97.13 (±0.23) | 96.07 (±0.10) | 95.42 (±0.17) | 98.93 (±0.14) | 98.88 (±0.19) | 98.86 (±0.11) | 98.77 (±0.09) |
| | 0.5 | 90.99 (±0.38) | 97.59 (±0.20) | 97.07 (±0.63) | 95.87 (±0.10) | 94.62 (±0.94) | 98.78 (±0.15) | 98.84 (±0.16) | 98.74 (±0.30) | 94.36 (±1.13) |
| | 0.7 | 81.85 (±0.77) | **97.11 (±0.15)** | 96.30 (±0.36) | 90.95 (±0.22) | 88.88 (±0.28) | 98.31 (±0.38) | **98.62 (±0.23)** | 96.67 (±0.14) | 90.51 (±1.09) |
| ResNet20-CIFAR10 | 0 | **91.76 (±0.28)** | - | - | - | - | - | - | - | - |
| | 0.3 | 90.98 (±0.23) | 89.15 (±0.51) | 90.13 (±0.95) | 90.67 (±0.49) | 90.84 (±0.43) | 88.69 (±0.80) | 89.54 (±0.57) | 90.30 (±0.26) | 90.15 (±0.31) |
| | 0.5 | 89.72 (±0.49) | 89.93 (±0.52) | 90.50 (±0.74) | 90.33 (±0.72) | 89.97 (±0.62) | 88.21 (±0.50) | 89.55 (±0.73) | 89.98 (±0.98) | 89.65 (±1.10) |
| | 0.7 | 82.03 (±0.76) | **88.88 (±0.74)** | 87.95 (±0.13) | 87.21 (±0.79) | 85.80 (±0.88) | 88.45 (±0.90) | **89.15 (±1.16)** | 88.52 (±1.29) | 87.60 (±1.41) |
| MLP-IMDB | 0 | **85.53 (±0.18)** | - | - | - | - | - | - | - | - |
| | 0.3 | 85.42 (±0.30) | 85.85 (±0.63) | 85.49 (±0.17) | 84.72 (±0.58) | 85.47 (±0.03) | 85.49 (±0.33) | 85.54 (±0.55) | 85.64 (±0.51) | 85.21 (±0.74) |
| | 0.5 | 84.85 (±0.63) | 85.44 (±0.68) | 85.35 (±0.84) | 84.06 (±0.58) | 84.55 (±0.72) | 85.55 (±0.69) | 86.00 (±0.36) | 85.18 (±0.62) | 85.92 (±1.22) |
| | 0.7 | 64.91 (±1.71) | 84.00 (±0.38) | **84.24 (±0.94)** | 82.83 (±0.36) | 80.90 (±0.71) | **85.11 (±0.40)** | 85.08 (±0.30) | 83.27 (±1.38) | 84.88 (±1.03) |
| LSTM-Names | 0 | **81.24 (±0.25)** | - | - | - | - | - | - | - | - |
| | 0.3 | 82.31 (±0.81) | 83.76 (±0.58) | 82.35 (±0.27) | 81.17 (±0.31) | 80.51 (±0.59) | 80.52 (±0.36) | 82.05 (±0.40) | 81.63 (±0.08) | 82.23 (±0.83) |
| | 0.5 | 56.91 (±1.42) | 82.17 (±0.64) | 81.70 (±0.78) | 81.56 (±0.45) | 81.43 (±0.95) | 80.13 (±0.52) | 82.54 (±1.03) | 82.04 (±1.21) | 82.57 (±0.06) |
| | 0.7 | 47.65 (±1.97) | **81.56 (±0.34)** | 80.87 (±0.58) | 81.07 (±0.81) | 66.68 (±0.57) | 79.35 (±0.35) | **81.15 (±0.75)** | 80.40 (±0.75) | 46.59 (±1.33) |

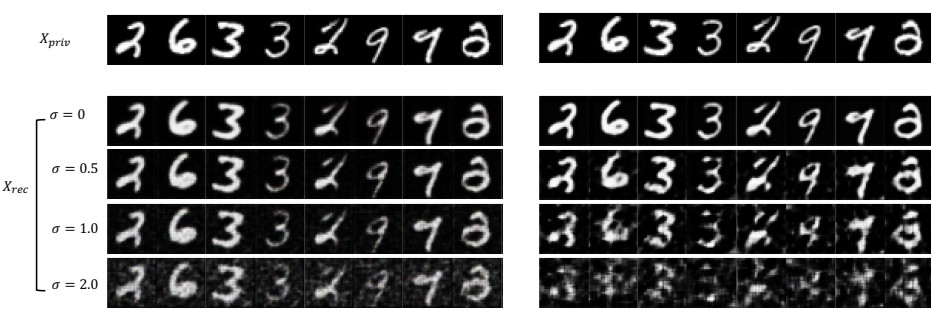

(a) 2 FC layers on server side, split layer size = 12544          (b) 1 FC layer on server side, split layer size = 256

Figure 9: Private data recovery by FSHA in split learning on MNIST. $X_{priv}$: the original private data, $X_{rec}$: obtained by FSHA attack. We compare two different split learning settings: (a) split 2 FC layers on the server side (b) split 1 FC layer on the server side. The dimension of the split layer are also different. We can see that by splitting more layers on the server side, it is more likely to reconstruct private training even when the noise level is relatively high ($\sigma = 1.0, 2.0$).

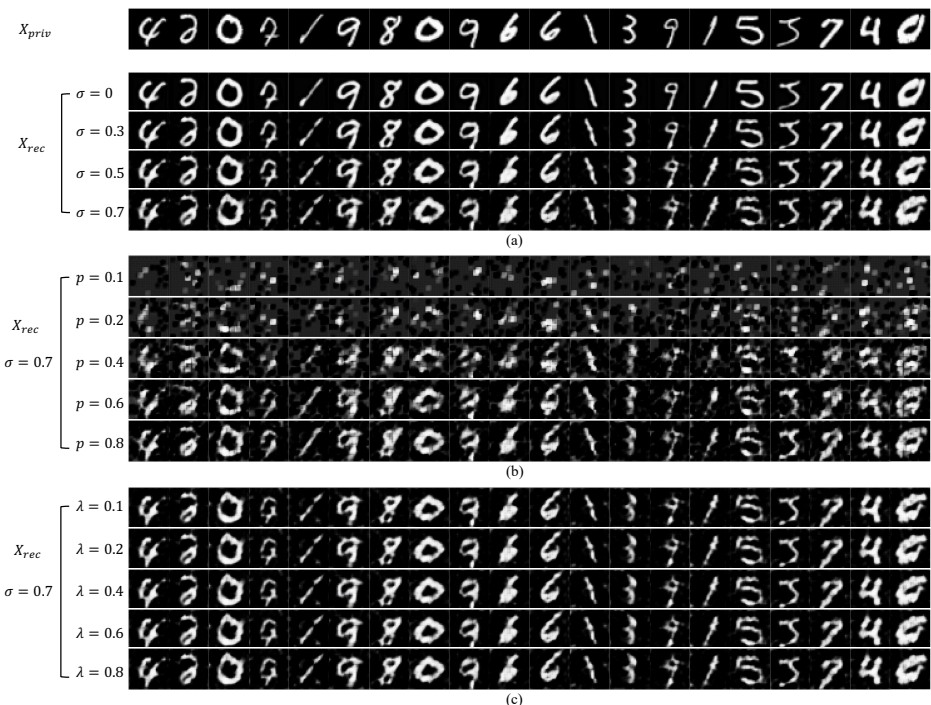

Figure 10: Private data recovery by FSHA in split learning on MNIST. $X_{priv}$: the original private data, $X_{rec}$: obtained by FSHA attack in various settings: (a) noise injection only (b) noise injection + masking (c) noise injection + scaling.

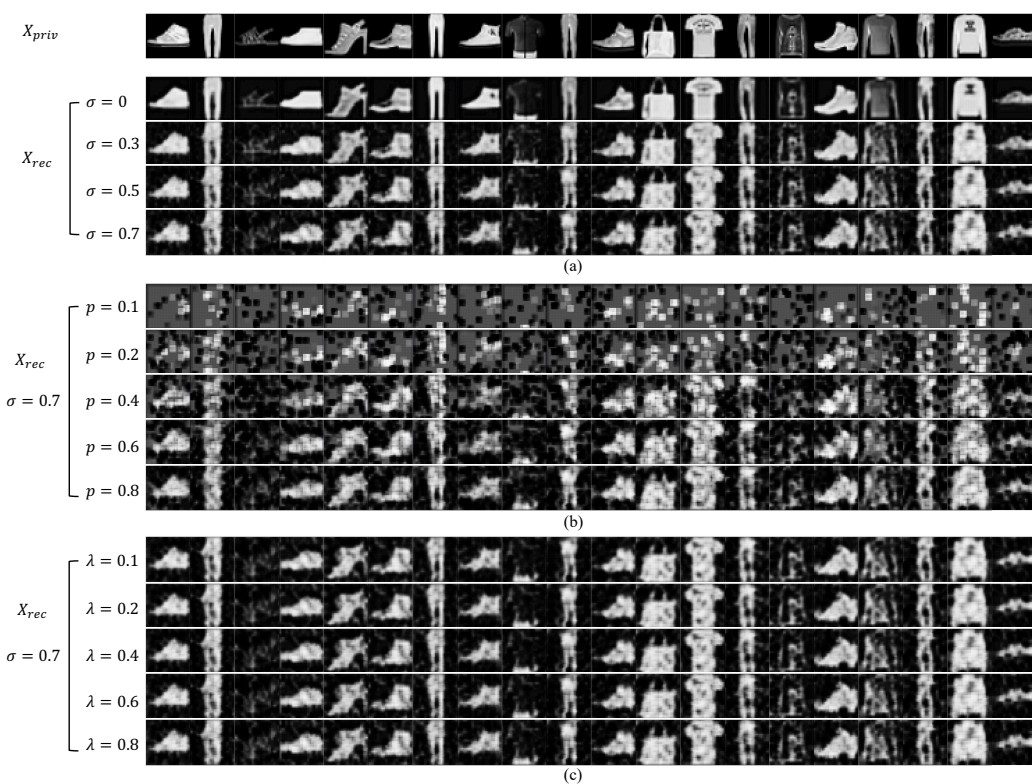

Figure 11: Private data recovery by FSHA in split learning on Fashion-MNIST. $X_{priv}$: the original private data, $X_{rec}$: obtained by FSHA attack in various settings: (a) noise injection only (b) noise injection + masking (c) noise injection + scaling.

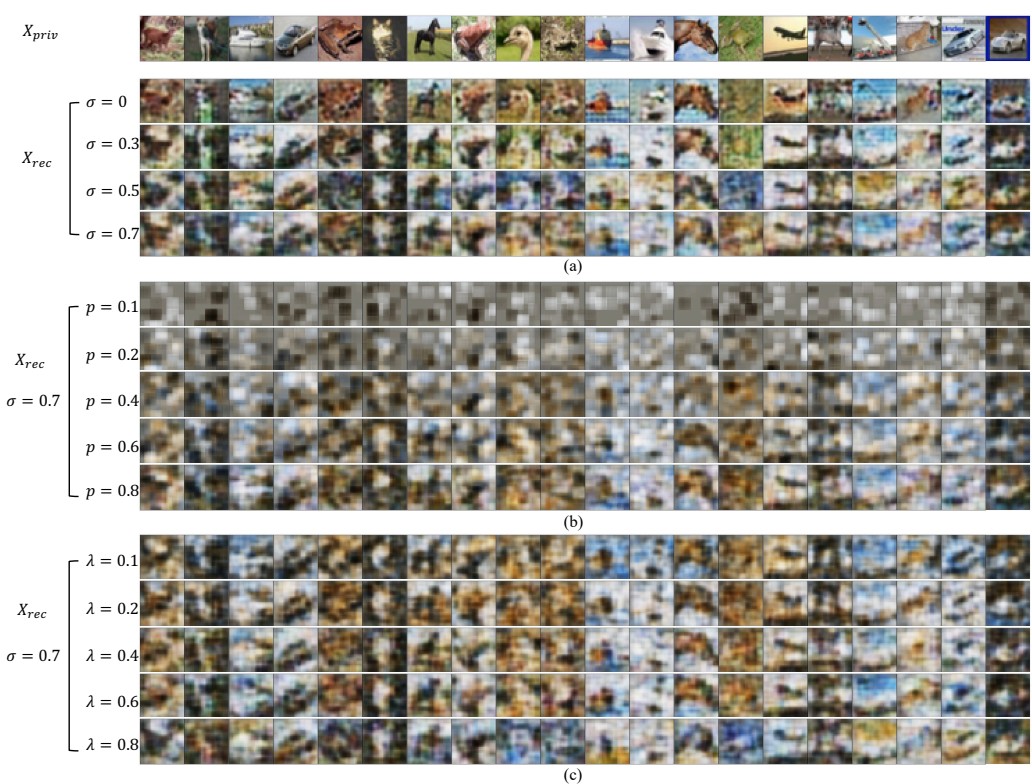

Figure 12: Private data recovery by FSHA in split learning on CIFAR-10. $X_{priv}$: the original private data, $X_{rec}$: obtained by FSHA attack in various settings: (a) noise injection only (b) noise injection + masking (c) noise injection + scaling.

Table 11: We provide the privacy bounds for one single forward pass during the SplitNN training, for various noise levels used in our work, without denoising. We compare it with two state-of-the-art results for private split learning.

| Method | Model & Dataset | DP mechanism | Inference | Training | Noise scale | DP bounds |
|---|---|---|---|---|---|---|
| Titcombe et al. 2021 | 2D CNN on MNIST | Laplace | ✓ | | 0.1, 0.5, 1.0 | N/A |
| Abuadbba et al. 2020 | 1D CNN on medical data | Laplace | | ✓ | N/A | $\epsilon = 1,3,5,7,10$ |
| Our work (w/o sampling amplification) | 2D CNN on MNIST | Gaussian | | ✓ | 0.3, 0.5, 0.7 | $\epsilon = (260, 150, 110)*2, \delta = 1e-5$ |
| | ResNet on CIFAR10 | Gaussian | | ✓ | 0.3, 0.5, 0.7 | $\epsilon = (264, 152, 112)*2, \delta = 1e-5$ |
| | MLP on IMDB | Gaussian | | ✓ | 0.3, 0.5, 0.7 | $\epsilon = (168, 96, 72)*2, \delta = 1e-5$ |
| | LSTM on Names | Gaussian | | ✓ | 0.3, 0.5, 0.7 | $\epsilon = (9.2, 5.3, 3.9)*2, \delta = 1e-5$ |
| Our work (w/o sampling amplification) | 2D CNN on MNIST | Laplace | | ✓ | 0.3, 0.5, 0.7 | $\epsilon = (53.3, 32, 22.8)*2$ |
| | ResNet on CIFAR10 | Laplace | | ✓ | 0.3, 0.5, 0.7 | $\epsilon = (53.3, 32, 22.8)*2$ |
| | MLP on IMDB | Laplace | | ✓ | 0.3, 0.5, 0.7 | $\epsilon = (13.3, 8, 5.7)*2$ |
| | LSTM on Names | Laplace | | ✓ | 0.3, 0.5, 0.7 | $\epsilon = (53.3, 32, 22.8)*2$ |

local SGD training using Theorem A.10 would result in a large privacy bound and proposed a new accounting for local noisy SGD training. However, noisy SplitNN is a more complicated architecture, and DP-SGD analysis cannot be adopted here. Currently, privacy accounting for noisy SplitNN training remains an open problem. Improving the privacy accounting for split learning is not the primary focus of this work, instead, we want to show how denoising can improve noisy SplitNN accuracy.

## C  LIMITATIONS

Our denoising techniques work empirically on diverse datasets (MNIST, FMNIST, CIFAR-10, CIFAR-100, ImageNet1K, IMDB, Amazon Reviews, and Names) and across different network architectures (CNN, RNN, Transformer, and MLP). One of the potential drawbacks or limitations of the denoising techniques empirically is finding a good scaling or masking ratio. Another potential limitation could be that our proposed denoising techniques may not work for other loss functions. However, theoretically, we covered almost all the existing loss functions used for common DNN training; please see our discussion in the Appendix. Generalizing our theoretical claims to a broader class of nonlinear loss functions, such as sparse categorical cross-entropy, which is also rarely used in practice, requires further non-trivial investigation and is a scope for future research. Our present theoretical analyses are in Section 3.1. and 3.2 consider the split layer at the pre-final layer of an $L$ layer DNN; analysis of the split at an arbitrary $i$-th layer, along with the final loss function used for DNN training, requires much more mathematical rigor.

