# OpenReview forum: "Accurate Split Learning on Noisy Signals"
_ICLR.cc/2026/Conference — ICLR 2026 Conference Withdrawn Submission_

### Official Review · Reviewer_zrUB · 2025-10-24

**Soundness:** 2
**Presentation:** 2
**Contribution:** 2
**Rating:** 2
**Confidence:** 4

**Summary:**

This paper proposes applying denoising techniques after injecting noise into intermediate results in split learning to improve overall performance. It provides a detailed theoretical analysis and evaluates the defense effectiveness against reconstruction attacks.

**Strengths:**

1. Proposes an effective approach to mitigate the adverse effects of noise injection.
2. Provides a detailed theoretical proof.

**Weaknesses:**

1. The method is relatively simple and appears to be a combination of noise injection with dropout or a variant of differential privacy, as DP also involves noise and scaling.

2. The masking ratio or scaling factor is difficult to determine given the noise ratio, making it challenging to apply in real-world scenarios.

3. All split layers are placed before the fully connected layers, which is unrealistic. In practical settings, the client typically holds only a small portion of the model—usually just a few layers—due to limited computational resources. Therefore, more diverse split configurations should be evaluated for both accuracy and attack performance, especially when assessing the defense.

4. The evaluated FSHA attack is outdated; more recent attacks such as PCAT [1], FORA [2], and SDAR [3] should be included. These attacks do not interfere with the split learning process and typically train a pseudo-client. The paper should also examine how these attacks adapt to noise (i.e., adaptive attacks) and how the defense performs under such conditions.

5. The defense evaluation is insufficient; more comparisons with existing defenses are needed, and the privacy–utility trade-off should be clearly demonstrated.


[1] Gao, Xinben, and Lan Zhang. "{PCAT}: Functionality and data stealing from split learning by {Pseudo-Client} attack." 32nd USENIX Security Symposium (USENIX Security 23). 2023.

[2] Xu, Xiaoyang, et al. "A stealthy wrongdoer: Feature-oriented reconstruction attack against split learning." Proceedings of the IEEE/CVF conference on computer vision and pattern recognition. 2024.

[3] Zhu, Xiaochen, et al. "Passive inference attacks on split learning via adversarial regularization." NDSS 2025.

**Questions:**

1. What is the difference between dropout and masking, and why do they have different effects on training?

---

### Official Review · Reviewer_Sxg4 · 2025-10-31

**Soundness:** 3
**Presentation:** 2
**Contribution:** 2
**Rating:** 4
**Confidence:** 3

**Summary:**

This paper discusses denoising schemes when the intermediate representations of a neural network was injected noise during training in a split learning setting. The authors propose to use scaling and masking to reduce the variance of the noisy output and theoretically shows that when a linear map and a classification loss are operated on the noisy intermediate representations, there exist parameter choices such that doing so would denoise the final output. The authors conduct experiments to verify the theory and extrapolate it on deeper networks.

**Strengths:**

1. Interesting perspective on how to train the network with noisy intermediate representations.
2. Theory seems sound and shows after transformation, the output/loss on the denoised intermediate representations is closer to the output/loss on the actual output/loss than using the noisy representations directly.
3. Experiments show that scaling and masking are indeed useful across different model and data scales.

**Weaknesses:**

1. The organization of this paper could be improved. Some important descriptions and explanations are deferred in the appendix while I believe should be clarified in the main text. For example, the definitions of the scaling/masking operators only appear in the appendix. This is the major algorithm proposal of the paper, and I believe that it should have been in the main paper. Also, I don't find the DP preservation part that necessary in the main text. Post-processing is a well known fact, while this paper mainly deals with the denoising part and do not care that much about what kind of guarantees the noise injection provides.
2. The main theorems seem not informative enough. Please see questions below.
3. The paper does not provide a principled way to choose the parameters in the proposed operators. Please see questions below.
4. DP analysis/discussion is not very clear and is not explained in the experiment section. Please see questions below.
5. Insufficient literature review on more recent attacks and defenses. The reference list seems to have a cut off year of 2023. There are many more recent works on SL privacy in recent years.

**Questions:**

1. Theorem A.1, 3.1 and theorem 3.2 all shows that the distance to the ground truth output or loss will be smaller if we use scaling or masking. However, this is not informative enough. For example, is it possible to bound the ratio between the distances tighter than 1? The experimental results seem very encouraging for that.
2. The theorems only state the existence of the scaling and masking parameters that achieves lower expected distance to the true output/loss, but do not describe how to find the optimal values. It shows that as long as certain conditions are met, we will have lower MSE. Different choices of the parameters still decide how much improvement we could have. Is it at all possible to find the optimal parameters by optimizing the distance ratio? From (11) and (13), it seems that with some assumptions on X, we can optimize these values?
3. At a result, the experiments do not explain the choice of the scaling and masking parameters, but instead show the improvement across different choices. How are these parameters decided? How are the other hyperparameters chosen in the experiments?
4. I understand that the proposed denoising scheme will not break the DP guarantees as a post-processing step. However, in the experiments section, are there any DP guarantees for the choices of $\sigma$? If so, the authors should provide them.
5. If we add noise to the activations and we want to protect the input features from being reconstructed, my understand is that we would like a LDP-like guarantee. In section 3.3, the authors seem to claim that doing denoising on noisy intermediate representations gives DP-SGD-like noisy gradients. My understanding is that DP-SGD-like differentially private model training is not applicable to defend against activation-based reconstruction attacks. Even if the network f is $\epsilon$-DP for very small $\epsilon$, that doesn't mean that $f(X)$ can not be used to invert $X$. Please clarify. Again, I do believe that this is not the focus of this paper, as the paper is mainly interested in improving the utility when noise is present in the intermediate presentations, regardless whether the noise provides meaningful DP guarantees.
6. Please conduct a more comprehensive review on recent related works.

---

### Official Review · Reviewer_ANdq · 2025-11-01

**Soundness:** 3
**Presentation:** 3
**Contribution:** 3
**Rating:** 4
**Confidence:** 3

**Summary:**

This work aims to improve the accuracy of models using noise injection in Split Learning as a privacy leakage mitigation. They do so by denoising using scaling and random masking. They supply theoretical results to back up their methods and present experimental results.

**Strengths:**

- The paper is well written and presented.
- Strong theoretical results, especially the differential privacy guarantees.
- Sound ideas.

**Weaknesses:**

The presentation and analysis of the empirical results are insufficient. Specifically:

- There is a lack of metrics to support the statement that noise injections combined with your denoising methods indeed are privacy preserving across the whole dataset. I would recommend in section 4.3 to not only compare the visualizations in a subjective manner as to avoid cherrypicking. You can use direct measures such as MSE, PSNR, SSIM and calculate them across the dataset. Other metrics you could explore could be training a separate classifier on the original data and using the accuracy of that model on your reconstructed dataset as a metric (lower being better).

- The paper would benefit from a table or figure showing test accuracy and reconstruction metrics across datasets and noise levels, both with and without the proposed denoising methods. This would strengthen confidence that the approach consistently improves the privacy-utility tradeoff between model accuracy and privacy protection

**Questions:**

Have you considered any adaptive attack version of FSHA that utilizes the knowledge that you are using scaling and random masking to perform denoising?

---

### Official Review · Reviewer_yB5b · 2025-11-01

**Soundness:** 3
**Presentation:** 3
**Contribution:** 3
**Rating:** 6
**Confidence:** 3

**Summary:**

In split learning, injecting noise into intermediate representations (IRs) protects privacy but hurts accuracy. This paper adds a server-side post-processing denoiser on the already noised IR, reducing forward deviation and improving accuracy and stability, while preserving privacy via DP post-processing immunity. Masking also substantially weakens feature-space hijacking attacks.

**Strengths:**

1. Minimal, deployment-friendly change that preserves DP by construction.

2. Theoretical forward-pass improvements with simulations that match, plus broad experimental validation.

**Weaknesses:**

1. The proofs concern one-step forward deviation. However, end-to-end training effects under different optimization algorithms are validated empirically but not analyzed. This limits formal guarantees on convergence and generalization.

2. Optimal scaling factors and mask rates are workload-dependent. the paper doesn’t propose adaptive selection rules with guarantees. And while DP post-processing ensures the budget isn’t worsened, an end-to-end accounting over many rounds would make the privacy story tighter.

3. Interactions with structures like BN, residuals and attention and effectiveness against other IR-level attacks are not deeply charted. Without such mapping, deployment guidance across diverse models/threats remains partly heuristic.

**Questions:**

See weaknesses.

---

### Note · Authors · 2025-11-21

I have read and agree with the venue's withdrawal policy on behalf of myself and my co-authors.